# Implications of Salt Diapirism in Syn-Depositional Architecture of a Carbonate Margin-to-Edge Transition: An Example from Plataria Syncline, Ionian Zone, NW Greece

Ioannis Vakalas [1,2,*], Sotirios Kokkalas [3], Panagiotis Konstantopoulos [4], Constantinos Tzimeas [5], Isidoros Kampolis [1], Helen Tsiglifi [6], Ruben Pérez-Martin [7], Pablo Hernandez-Jiménez [7] and Juan Pablo Pita-Gutierrez [7]





1    Department of Geological Sciences, School of Mining and Metallurgical Engineering, National Technical University of Athens, Iroon Polytechniou 9 Str., Zografou Campus, 15773 Athens, Greece; kampolisigeo@gmail.com
2    Institute of GeoEnergy, Technical University of Crete Campus, 73100 Chania, Greece
3    Department of Geology, University of Patras, 26504 Patras, Greece; skokalas@upatras.gr
4    Energean Oil & Gas, 32, Kifissias Avenue, Atrina Center, 15125 Marousi, Greece; pkonstantopoulos@energean.com
5    Freelance Exploration Geophysicist, 4 Platonos Str., 15121 Ano Pefki, Greece
6    Ionia Labs Soil & Rock Mechanics Laboratory, Nea Ionia, 14565 Athens, Greece; e.tsiglifi@yahoo.com
7    Repsol Exploración SA, Mendez Álvaro 44, 28045 Madrid, Spain; perez.martin.ruben@repsol.com (R.P.-M.); p.hernandez@repsol.com (P.H.-J.); jppitag@repsol.com (J.P.P.-G.)
*    Correspondence: ivakalas@metal.ntua.gr; Tel.: +30-2107723970

**Abstract:** The present study examines the imprint of salt tectonics on carbonate depositional patterns of the Ionian zone platform edge to slope transition. The study area is part of an overturned rim syncline adjacent to a salt diapir. The Ionian zone is made up of three distinct stratigraphic sequences (pre-, syn- and post-rift sequences) represented by evaporites and shallow water carbonates at the base that pass gradually to a sequence consisting of pelagic limestones with shale intervals. In the study area, six cross sections were constructed, mainly covering the edge-to-slope overturned succession of Early Cretaceous to Eocene carbonates (post-rift stage) in the northern limb of the syncline. In the measured sections, abrupt changes in sediment texture resulted in the formation of distinct, thick-bedded carbonate layers, identified as packstones to grainstones–floatstones, with abundant fossil fragments, indicating deposition by debrites in a platform slope or slope-toe environment. Planar and ripple cross-lamination also suggest the involvement of turbidity currents in the depositional process. In the upper levels of the Lower Cretaceous carbonates, chert bodies with irregular shapes indicate soft sediment deformation due to instability of the slope triggered by salt intrusion. Internal unconformities identified in the field and in the available seismic data combined with the vertical to overturned dipping of the strata correspond to a basal megaflap configuration. Syn-sedimentary deformation resulted in the accumulation of debritic and turbiditic layers, while the compressional regime established in the area from the Late Cretaceous to Early Eocene enhanced the fracture porosity of carbonates, which could eventually affect the reservoir properties.

**Keywords:** evaporite intrusion; reactive piercement; platform-edge to slope transition; carbonate stratigraphic architecture; upturned flaps; basal megaflap

## 1. Introduction

Salt tectonics play a key role in hydrocarbon exploration worldwide (e.g., Gulf of Mexico, Campos Basin, Lower Congo Basin, Persian Gulf, North Sea), not only by favoring the formation of effective seals but also by controlling depositional processes and, subsequently, reservoir properties [1–3]. Additionally, the need for a reduction in anthropogenic

greenhouse gas emissions into the atmosphere drew attention to the possibility of geological storage of carbon dioxide in specific formations, including salts and salt structures [4,5]. Salt structures have been extensively discussed in the literature, regarding the mechanics of salt flow, diapiric growth and their effects on passive margin evolution [6–8]. Shallow and deep water systems of mixed siliciclastics and carbonates have been studied by many researchers [7,9–14] and, to a lesser degree, pure carbonates systems have also been studied [15]. Various salt-tectonic structures, such as salt pillows, salt anticlines, salt walls and stocks, salt sheets and canopies, are identified in outcrop and subsurface data. The mechanism and the driving forces for the formation of these structures and diapiric growth may involve extensional, contractional forces governed by the regional tectonic regime, or may be controlled by purely halokinetic processes. Salt pillows and anticlines can form by halokinesis, growing purely by gravitational forces, or either by contractional folding or in the cores of normal fault rollovers [3,16]. Three modes are involved in the formation of the aforementioned geometries: reactive, active and passive diapir piercement, depending on the regional tectonic regime [2,17–21]. Hudec and Jackson [22,23] distinguished four models describing the advance mechanism of salt sheets and canopies: (a) extrusive advance, where a salt spreads in the surface without any roof; (b) open toe advance, where a part of the salt is partially buried by a roof; (c) thrust advance, where a buried sheet advances along with its roof in the hanging wall of a thrust (common pattern in foreland basin settings, as with the one in the Ionian zone, Greece); and (d) salt wing intrusion, where a wedge of salt is emplaced along a stratigraphic horizon or thin layer in the flanking in situ rock.

In order to analyze the effect of salt uprise on the geometrical and sedimentological features of the overburden, the regional stratigraphic sequences can be distinguished in three categories: (a) pre-kinematic sequences, (b) syn-kinematic sequences and (c) post-kinematic sequences. In salt pillows and anticlines, pre-kinematic sequences tend to maintain their initial thickness, while syn-kinematic sequences are thin close to salt crests forming on lap structures with the underlying pre-kinematic sequences [24]. According to Escosa et al. [25], strata geometries close to salt walls range from halokinetic sequences to megaflaps. Halokinetic sequences are described as localized (<1 km wide) successions of growth strata that form as drape folds. The controlling factor of their geometry is the interplay between the salt-rise rate and the sediment-accumulation rate [26,27]. Stacked halokinetic sequences form tabular and tapered composite halokinetic sequences [28]. Megaflaps have been defined as panels of deep minibasin strata that extend far up the sides of steep diapirs or their equivalent welds [6,29,30]. In all cases, the post-kinematic sequences are marked by unconformity surfaces pointing at the cease of salt uprise.

The present study focuses on a key area of NW Greece, since it forms part of an overturned 'rim syncline' adjacent to a salt diapir wall. Extensional and compressional tectonics evolved in the area during the geological record, and combined with the presence of a thick evaporitic succession established a complex configuration that involved reactive and active diapirism. The role of salt tectonics, though partially obscured by the reactivated compressional stage, played a key role in the stratigraphic architecture of the carbonate margin to slope transition and its structural evolution.

## 2. Geological Setting

The study area was located in the Ionian zone, which forms part of the External Hellenides in Western Greece (Figure 1). The Ionian zone has been extensively studied, as it has attracted for more than half a century the interest for hydrocarbon exploration. Despite that, there is still much ambiguity in the deeper structure of the crust. The External Hellenides are considered mainly as either a region of thin-skinned tectonics or displaying a coexistence of thin- and thick-skinned deformation of the Mesozoic-Cenozoic succession above the Triassic evaporites and Permian basement sequences [31–34]. Structural [33–36], sedimentologic-stratigraphic [37–43], geochemical and paleontological studies [44–52] cover almost the whole extent of the exposed stratigraphy in an effort to evaluate the

hydrocarbon potential of the Ionian zone [53–56], not only in Western Greece but also in Albania [57–59] and in similar settings in Italy [60,61].

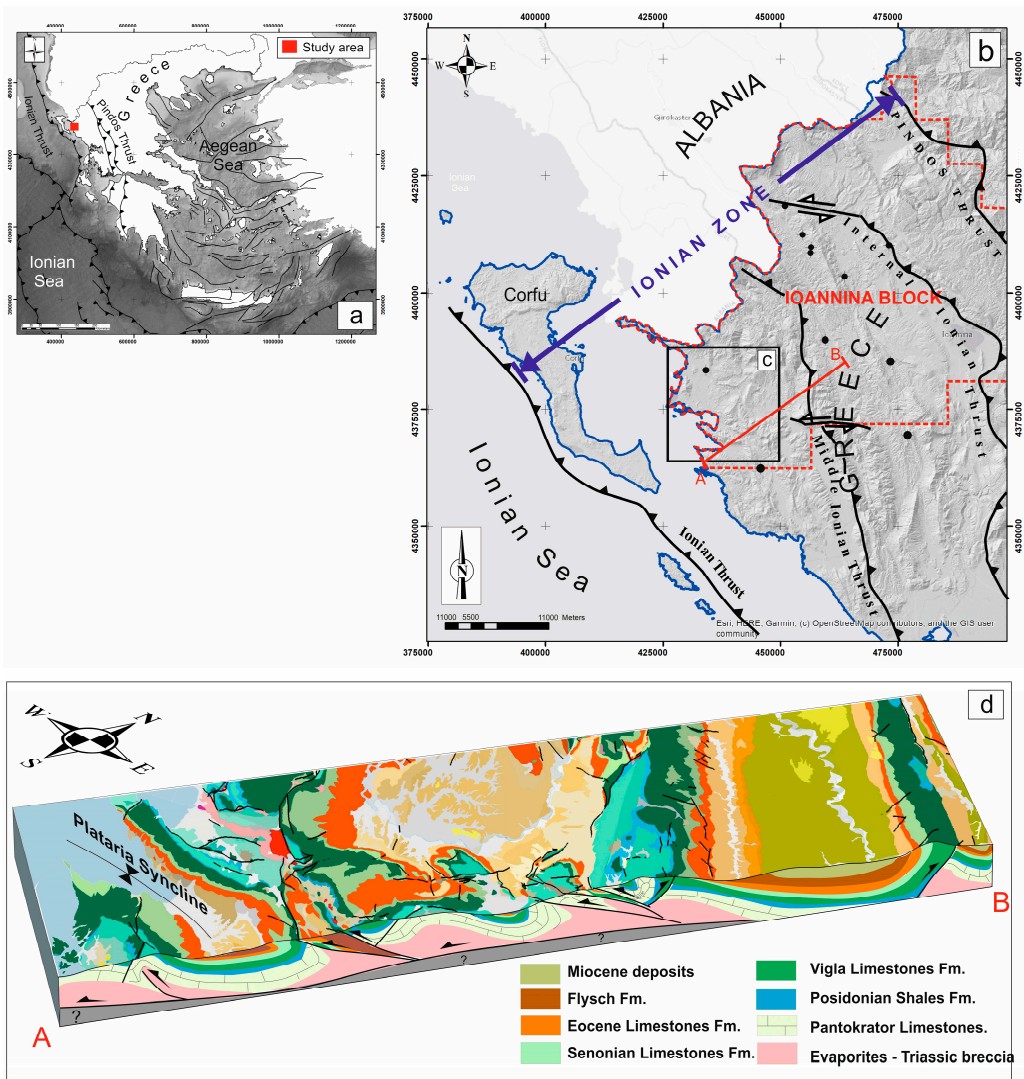

**Figure 1.** (**a**) Inset map of Greece with the main structural features and the location of the studied area highlighted with a red square. (**b**) Reference map showing the extent of the Ioannina concession (red dashed line). Well locations of previous exploration projects are also shown (black dots), while inset black rectangle (**c**) shows the extent of the geological map presented in Figure 2. The spatial extent of the Ionian zone is defined by Pindos and Ionian thrust, respectively. (**d**) Block diagram map showing the surface geology of the broader area and representative schematic cross sections (A,B) showing the general salt-related, fold-thrust deformation style in the Ionian zone. The location of the cross section is shown in (**b**).

Mesozoic facies units in the Hellenides orogenic belt were deposited on a series of platforms (Pre-Apulian and Gavrovo zones) and deep basins (Ionian and Pindos zones) that formed the eastern rifted margin of the Apulian plate, bordering towards the east the Pindos Ocean [62–64]. These units (External Hellenides), emplaced during Cenozoic times following the closure of the Pindos ocean and the consequent continent–continent collision between the Apulia and Pelagonia microcontinents to the east, resulted in the inversion of Mesozoic basins, forming a series of thrust sheets [32,65,66].

The Ionian zone consists of sedimentary rocks ranging from Triassic evaporites to Jurassic–upper Eocene carbonates and minor cherts and shales (Figures 2 and 3). Thick Upper Eocene [67] to Miocene flysch deposits [68] overlie the aforementioned deposits. The

Ionian zone boundary towards the west is marked by intrusive evaporites that represent the lowest detachment level of individual overthrust sheets in the external Hellenides [69]. Contractional deformation was so intense that it overprinted the halokinetic movements that occurred during Mesozoic extension. Many researchers support the idea of a moderate to major evaporitic décollement level rather than a widespread diapirism [70–74], proposing a structural mechanism of the evaporites similar to that in thin-skinned fold-and-thrust belts in western Europe [75–78].

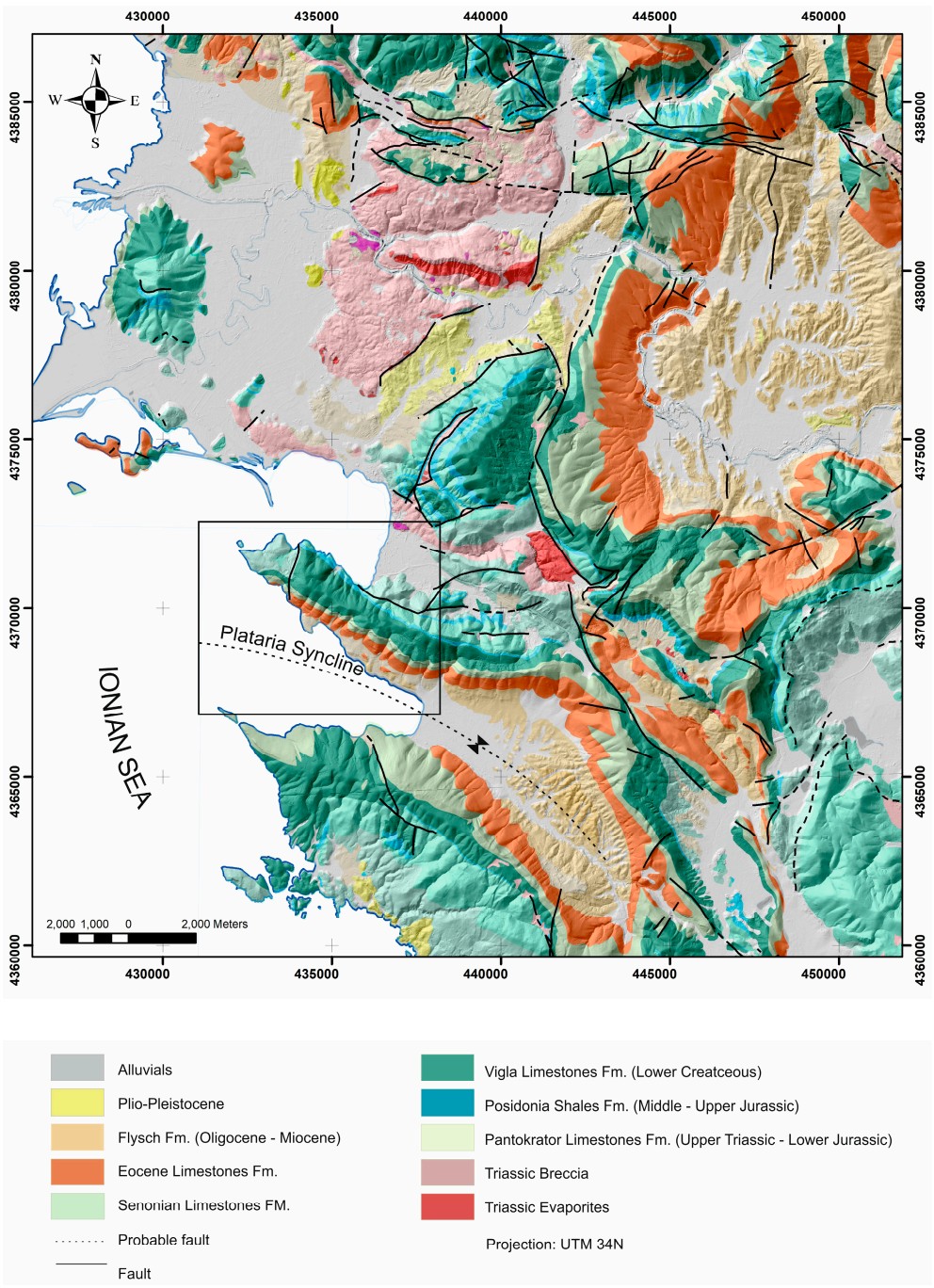

**Figure 2.** Geological map of the wider region of Plataria syncline. Notice the extension of Triassic Evaporites and Triassic breccia. The location of the northern limb, where the inversion of the stratigraphic succession occurs, is also shown (black rectangle). The geological map has been modified using the geological maps of the Institute of Geology and Mineral Exploration—IGME, in scale 1/50,000, Filiates, Parga and Sayiadha [79–81].

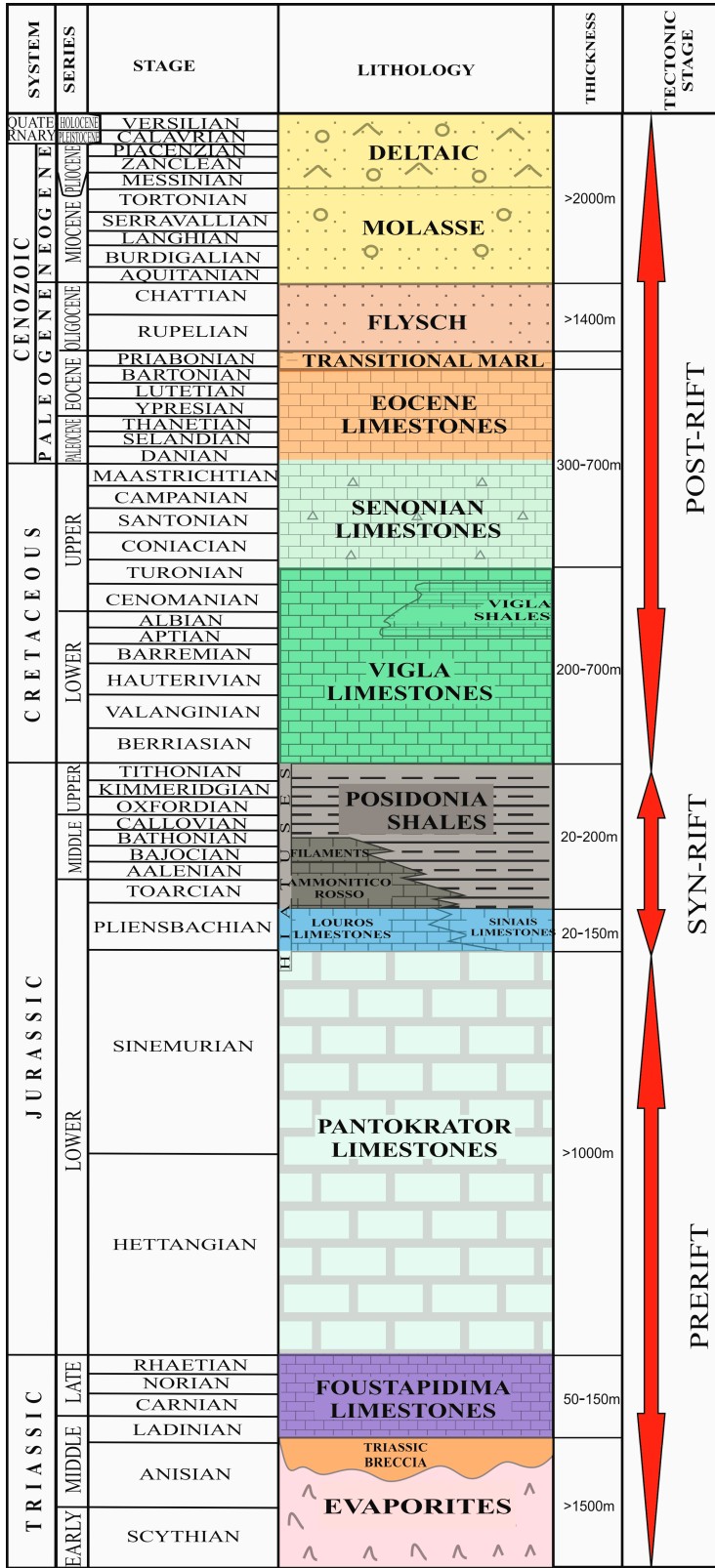

**Figure 3.** Lithostratigraphic column of the Ionian zone (modified after [1,67]).

According to the existing models, the Ionian zone consists of three distinct stratigraphic sequences [71,72] (Figure 3): (a) a pre-rift sequence (Triassic—Early Jurassic) represented by a thick evaporitic succession overlain by shallow marine carbonates; (b) a syn-rift sequence (Early to Late Jurassic) where the basin was compartmentalized and a half-graben geometry

was favored, reflected in variations in sediment type (pelagic carbonates and deep water clastics) and thickness. During this stage, the evaporites started to mobilize, initiating the formation of diapirs [72]; (c) A post-rift sequence (Early Cretaceous to Eocene) marked by the deposition of pelagic limestones. However, new insights into the evolutionary structural model of the Ionian fold-and-thrust belt in central-south Albania and western Greece came to light recently, taking into account the significant amount of new seismic and fieldwork data acquired over the last decade in the context of oil and gas exploration. According to these data, various researchers [82–84] proposed an alternative model, placing the rifting stage earlier in the Triassic and presenting the clear influence of salt tectonics since the Early Jurassic.

The pre-rift sequence comprises shallow water carbonates that overlie thick evaporitic deposits (~2000 m). Dissolution breccia overlain by dark-colored limestones (Foustapidima formation) mark the transition from the evaporitic succession to platform carbonates. The syn-rift sequence initiates with pelagic and hemipelagic limestones of the Pliensbachian age (Sinais and Louros formations). The transition from shallow marine to hemipelagic and pelagic formations is indicative of the gradual deepening of the basin. Carbonates and siliciclastics are deposited in the specific stage, characterized by lateral variations in thickness. These variations portray both the extensional tectonic regime and the mobilization of the underlying evaporites, which resulted in the segmentation of the basin into smaller sub-basins presenting a half-graben geometry [72]. The post-rift sequence is marked by pelagic limestones (Vigla limestones) which have been deposited across the whole extent of the Ionian basin. Small-scale differentiations in Vigla formation thickness suggest that halokinetic events were still active [69]. Vigla limestones are overlain by calciturbiditic limestones (Senonian Limestones).

According to [85], the Ionian Basin at that time was characterized by a central shallower part, surrounded by slopes that displayed higher sedimentation rates. The Apulian and Gavrovo platforms acted as sources of carbonate clastic material, which was deposited in the Ionian basin via gravity flow processes (debris flows and turbidity currents). During the Eocene time, clastic material supply was reduced and pelagic limestones that resemble those of Vigla formation were deposited. The basic difference between the two pelagic facies is the absence of the extensive chert layers that occur in Vigla limestones.

In the early to middle Eocene, carbonate sedimentation was followed by a gradual passage to clastic sedimentation, expressed by thick submarine fan deposits [67]. The transition was marked by a zone of calcareous marls, reaching a maximum thickness of 40–50 m [67]. At this stage, the basin was transformed into a typical foreland setting [67,70,86–89] with a major bounding thrust towards the east (Pindos Thrust). Inversion tectonics resulted in the re-activation of Jurassic extensional structural elements. Reverse or strike-slip faults indicate the change in the tectonic regime to a compressional configuration. As a result, minor thrusts divide the Ionian zone into the internal, middle and external Ionian zone (from east to west) [85]. A major orogenic phase occurred at the end of the early Miocene [85].

The age of Pindos foreland sediments is still a matter of debate. British Petroleum [90] proposed an early Miocene to middle Miocene age, interpreting the presence of Oligocene fauna as a product of large-scale erosion and the reworking of older sediments during the Miocene. In contrast, the IFP [85] suggested a late Eocene to early Miocene age for the basin infill, while others assigned an Oligocene age [91–94]. According to Vakalas et al. [95], the internal Ionian zone deposits range from Early Eocene to Late Oligocene. Avramidis et al. [96] proposed a middle Eocene to early Miocene age, using nannofossil zones from three cross sections in the Klematia-Paramythia basin (middle Ionian zone).

The role of pre-existing evaporite diapirs is significant in the compressional stage. Seismic data were interpreted to indicate a moderate decollement along the evaporites [72,97], while recent research supports the hypothesis of a major decollement [69].

## 3. Methods

The present study aims to incorporate high-resolution field work data with the seismic profiles provided by Repsol S.A. and Energean Oil & Gas. The first part of the fieldwork comprised detailed facies analysis of the exposed formations in six cross sections (Figure 4a) that trend normal to bed strike. Structural data (Figure 4b) were acquired in each of these sections to assess the geometrical correlations of the exposed lithologies.

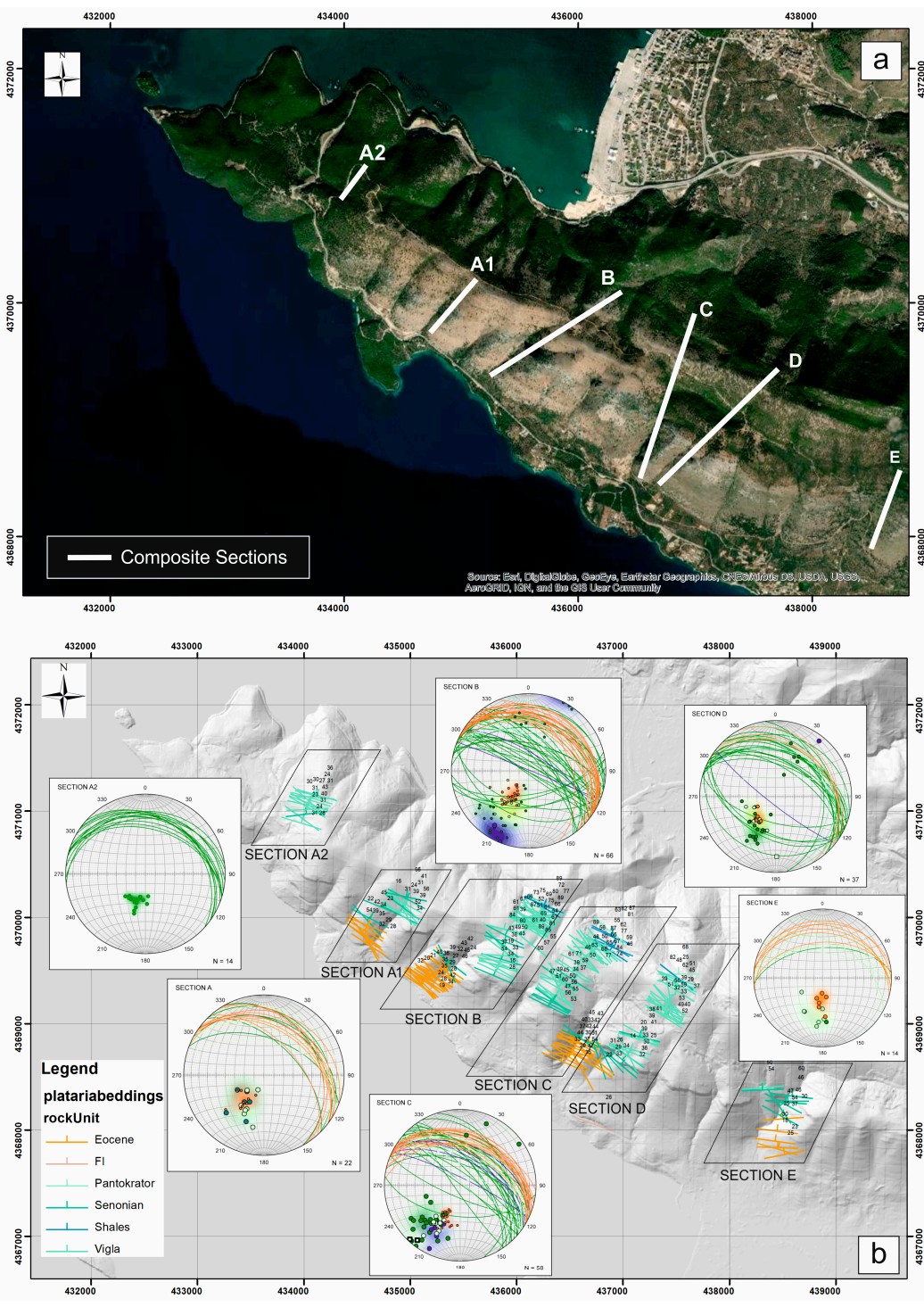

**Figure 4.** (**a**) Map of the study area showing the six cross sections' extent. (**b**) Relief map showing the structural measurements on stereo-diagrams across the cross sections. UTM grid spacing is 1 km.

The second task was to apply remote sensing techniques by capturing high-resolution aerial images in order to prepare 3D surface models. This process, combined with outcrop data, resulted in a better understanding of the lateral extent of the various lithostratigraphic units and their structural correlations.

Field data were then co-assessed with the available subsurface information, which consisted of one regional 2D depth-migrated, NE–SW trending, seismic reflection profile (see Section 6 for details).

## 4. Description of the Lithostratigraphic Columns

Six cross sections were examined (Figure 5) to identify the stratigraphic elements of the studied area and to construct the stratigraphic columns. Thickness measurements were made following the techniques proposed by [98,99]. Considering that the majority of the exposed rocks are carbonates, [100,101] classification schemes were applied.

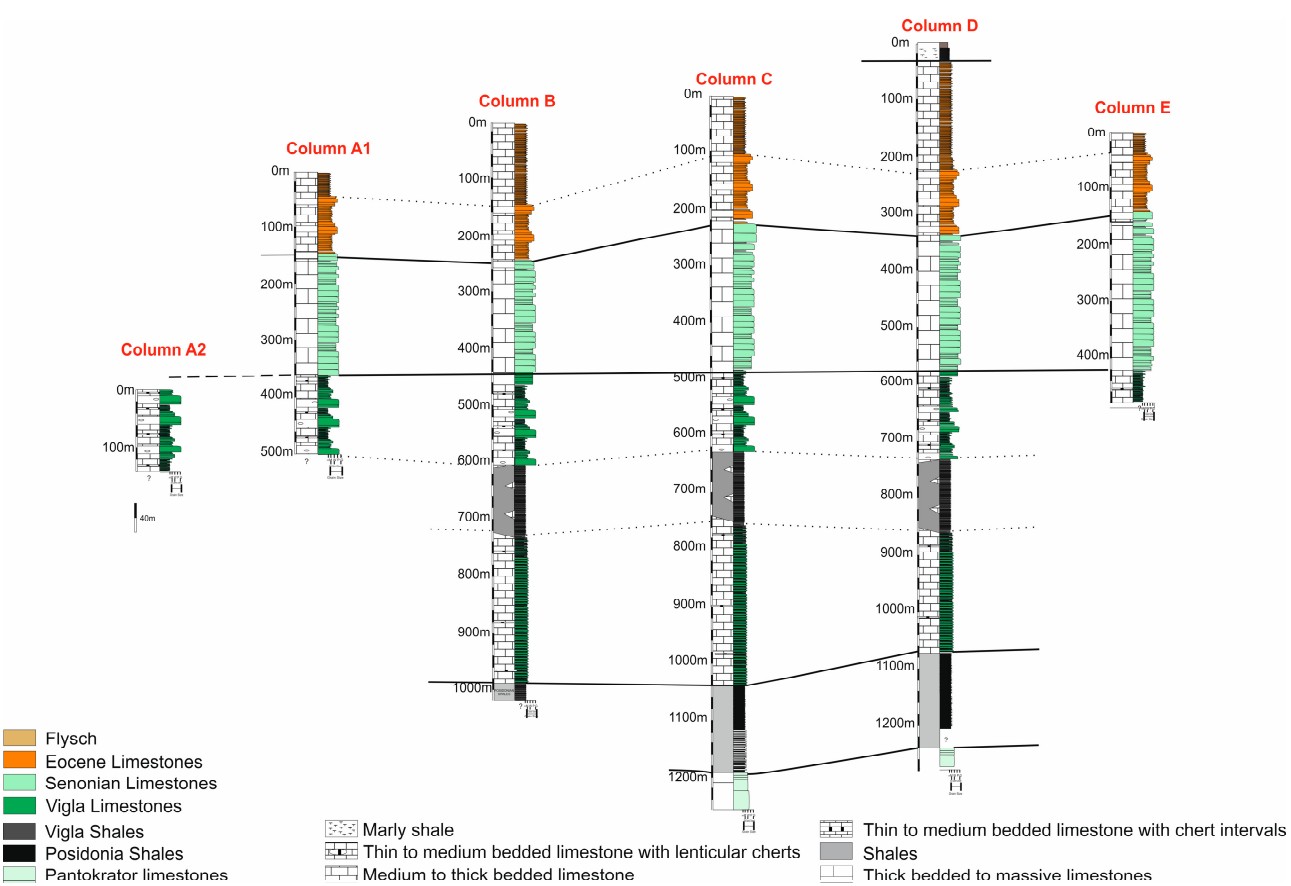

**Figure 5.** Synthetic lithostratigraphic columns of the measured sections in the studied area.

The whole succession, described below in the six cross sections, comprises part of the overturned limb of the Plataria anticline. Although in field sections the stratigraphy is inverted, the description of the formations in all sections is presented for consistency from top (Eocene) to bottom (Late Jurassic–Early Cretaceous) of the stratigraphic sequence.

### 4.1. Section A1

Section A1 starts with thin- to medium-bedded, creamy to white calcareous mudstones corresponding to Eocene limestone formation. Locally, fossiliferous (nummulites) horizons occur (Figure 6B). It is overlain by medium-bedded to massive limestones (Figure 6A). Medium-thickness beds are characterized by a mudstone texture, while thicker to massive beds present a wackestone to packstone texture. Cherts with nodular or irregular shapes are floating in the calcareous mass.

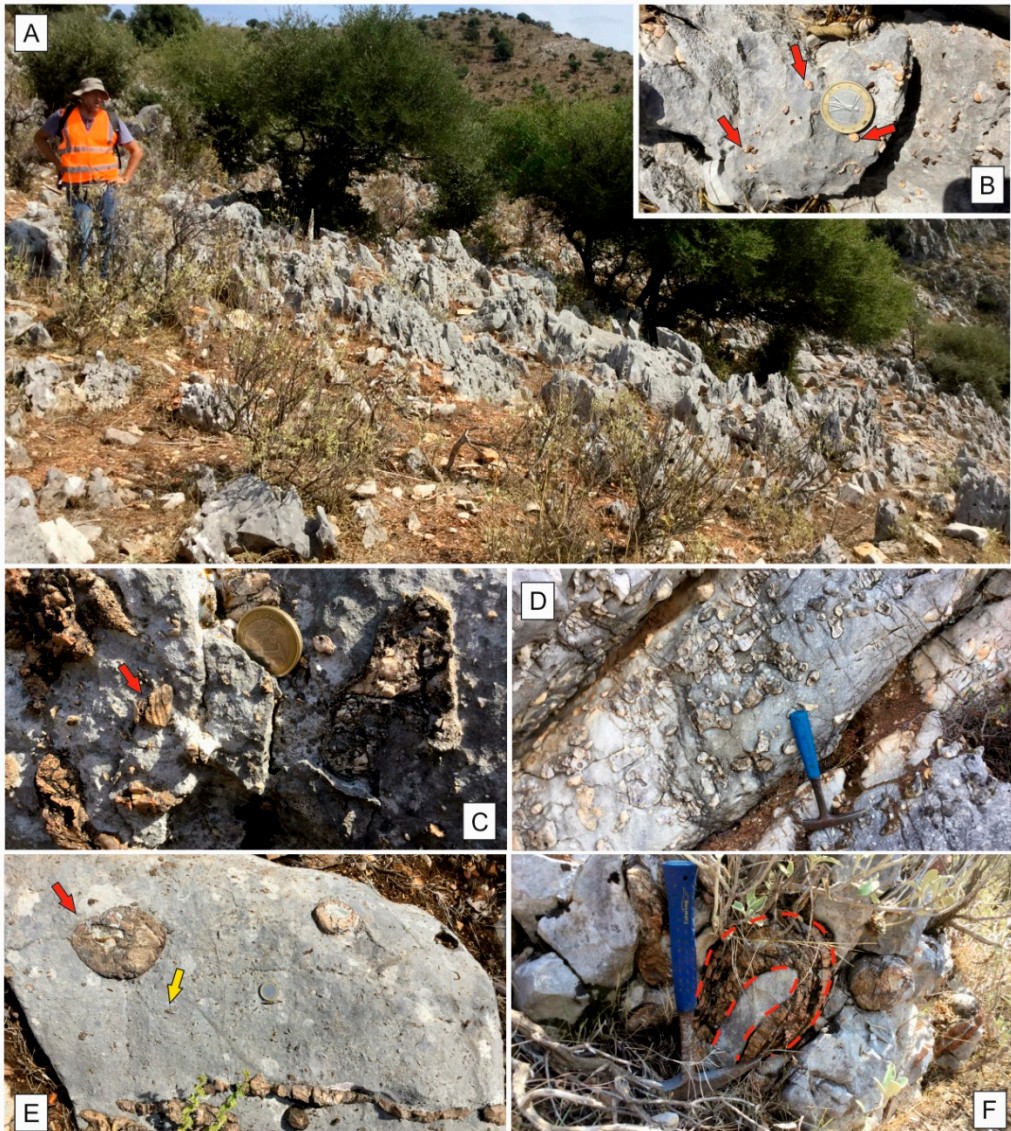

**Figure 6.** (**A**) Typical thick-bedded limestones of Senonian formation. Susceptibility to surface erosion is obvious. (**B**) Well-preserved nummulite species (red arrows) of the Eocene formation. (**C**) Fossil fragments (rudists) at the upper levels of Vigla limestones. (**D**) Intervals of chert layers in the Vigla limestones. The geometry of the layers implies soft sediment deformation. (**E**) Clasts (yellow arrow) of various sizes floating in a calcareous matrix, probably representing a calcidebrite of the Vigla formation. (**F**) Folded chert body (red dotted line), characteristic of soft sediment deformation.

Thick-bedded to massive packstones and wackestones (Senonian limestone formation) appear stratigraphically below the Eocene limestones and are characterized by the presence of scarce, thinner mudstones. Sedimentary features, such as planar or ripple cross-lamination, suggest the involvement of turbidity currents in the depositional processes. Fossil fragments including rudists are common.

The next formation (Vigla limestones) comprises thin- to medium-bedded limestones characterized by a mudstone texture with lenticular chert bodies trending parallel to bedding (Figure 6D). Alternations of thin- to medium-bedded limestones, characterized by a mudstone texture, with thick-bedded to massive limestones complete the measured stratigraphy. Thick to massive limestones resemble turbiditic deposits. Fossil fragments (Figure 6C) and cherts with irregular shapes are common (Figure 6F), suggesting soft sediment deformation. Wackestone to packstone texture is dominant (Figure 6E). Three horizons of massive limestones have been identified at the specific level of the Vigla

formation. From a geometrical point of view, onlaps are identified in various levels of the stratigraphy, implying that sedimentation took place while the basin was being uplifted.

### 4.2. Section A2

Section A2 focuses only on the lower Cretaceous limestones of the Vigla formation. The lower part of the measured stratigraphy consists of thin- to medium-bedded limestones characterized by a mudstone texture. These limestones are overlain by three intervals of thick- to massive-bedded limestones (ranging from 10 to 15 m in thickness) characterized by a packstone texture (Figure 7C) and sedimentary features, such as erosive surfaces (Figure 7D) and planar and ripple cross-lamination (Figure 6E), which indicate deposition by turbidity currents. Syn-sedimentary slumps (Figure 7A,B) suggest a depositional environment of unstable conditions (slope). Abundant fossil fragments and nodular cherts occur in the thick-bedded (Figure 7F,G) units, while the other stratigraphic intervals are layered.

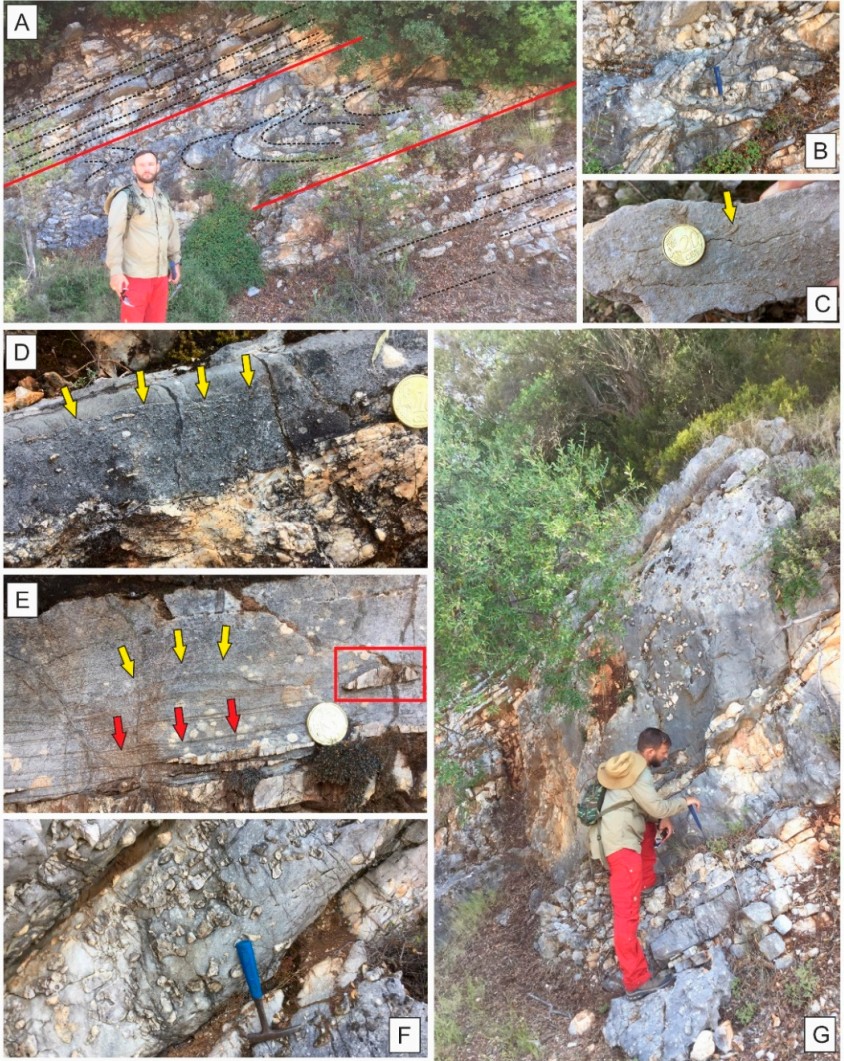

**Figure 7.** (**A**,**B**) Syn-sedimentary slump horizons in Vigla limestones. Red lines represent the bedding surface. (**C**) Packstone with various clasts floating in a calcareous matrix. (**D**) Well-formed erosive surface (yellow arrows), representing the contact between a poorly organized, matrix-supported debrite with a uniform mudstone (at the top). (**E**) Discrete ripple cross and planar lamination (yellow and red arrows, respectively) corresponding to Tc and Td Bouma sequence intervals. Notice the chert clasts that is in conformity with the cross and planar lamination surfaces (red square). (**F**) Chert clasts with irregular shapes floating in the limestone bed. (**G**) Massive limestone bed, typical for the upper Vigla unit in the study area.

### 4.3. Section B

Section B starts with thin- to medium-bedded, creamy to white calcareous mudstones of the Eocene limestone formation. Locally, fossiliferous (nummulites) horizons occur. Chert intervals characterized mainly by a lenticular geometry are also recognized. It is then succeeded by medium-bedded to massive limestones. Thinner beds are characterized by a mudstone texture, while thicker to massive beds present a wackestone to packstone texture. Sedimentary features expressed by planar and ripple cross-lamination suggest deposition by turbidity currents (Figure 8F). Cherts with nodular or irregular shapes are floating in the calcareous mass.

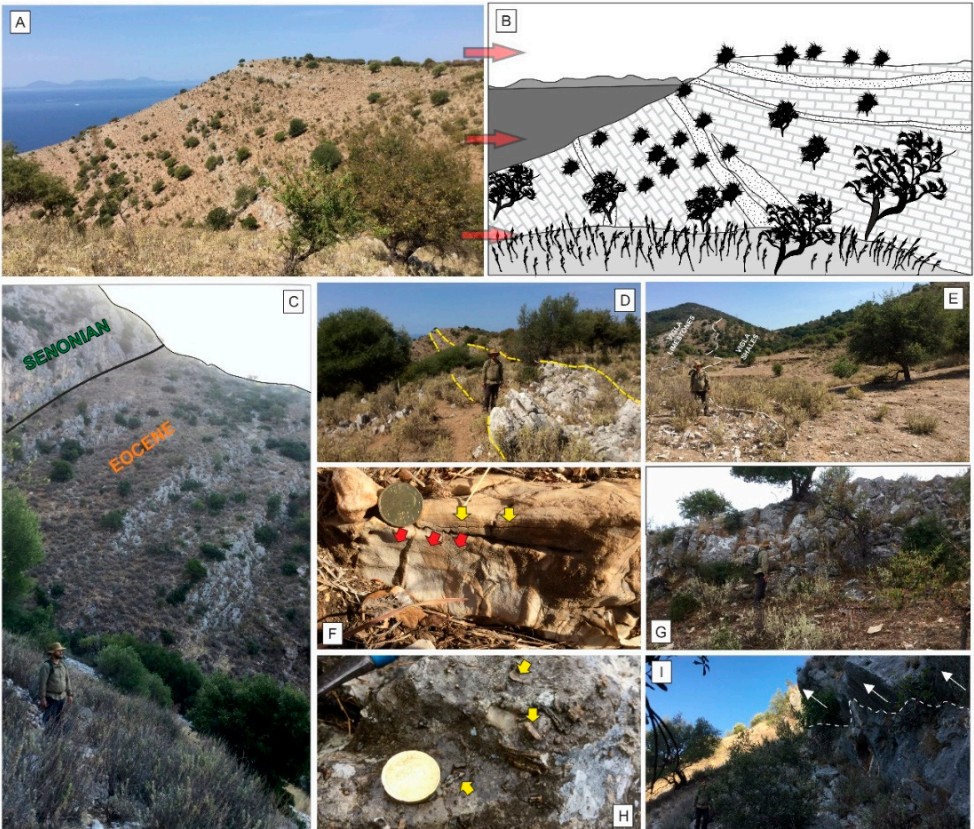

**Figure 8.** Photo (**A**) and sketch representation (**B**) of the upper unit of Vigla limestones, where internal onlaps are clearly identified. (**C**) Senonian massive limestones found topographically on top of Eocene formation limestones, due to inversion of the whole succession. A megaflap configuration with Senonian on top of Eocene can be considered as a possible setting. (**D**) Photo showing the lateral extent of Vigla calciturbidites. (**E**) Upper Vigla shales (left part of the photo) overlying lower Vigla Limestones. (**F**) Planar (yellow arrows) and ripple cross-lamination (red arrows) in medium-bedded limestones of Vigla formation. (**G**) View of the thick-bedded Vigla limestone calciturbidites. (**H**) Abundant fossil fragments floating in a calcareous matrix of Vigla limestones (yellow arrows). (**I**) Reverse fault (white arrows indicating probable fault movement) with a well-preserved fault surface (Senonian limestone formation).

Similar to section A1, the Eocene limestones overlay the Senonian limestones (Figure 8C), where thick-bedded to massive limestones are present. A packstone and wackestone texture is dominant, while some thinner beds are characterized by a muddy texture. Tc and Td Bouma sequence intervals expressed by current ripples and planar lamination indicate low or high density turbidites. Bedding planes are generally tabular, probably suggesting deposition in a slope-toe or basinal setting where the flows were less confined. Rudist fragments are widely abundant, floating in the calcareous matrix. At this level, low-angle reverse faults

were identified (Figure 8I), while a major normal fault, trending NW–SE, produces the steep scarp characterizing the Senonian formation across the Plataria mountainous area.

The Vigla formation starts with alternations of thin- to medium-bedded mudstone, interrupted by thick-bedded to massive wackestones to packstones, which resemble turbiditic deposits. Fossil fragments are also abundant at this level (Figure 8H). Similar to section A2, three horizons of massive limestones (Figure 8G) were identified in the specific level of the Vigla formation (Figure 8D). Internal onlaps are also common at the specific levels (Figure 8A,B). In section B1, a greater extent of the stratigraphy is exposed. Below the calciturbiditic/calcidebritic calcareous deposits, very thin- to thin-bedded, brownish to greenish, clayey shales alternate with brownish and greenish cherts. These deposits are very poorly exposed, but their spatial extent is marked by a smooth topographic relief (Figure 8E).

Below the Vigla shales, thin- to medium-bedded, creamy to light grayish calcareous mudstones with chert intervals mark the base of the Lower Cretaceous Vigla formation.

The measured section ends with poorly exposed thin-bedded shales with intervals of thin-bedded limestones corresponding to the Jurassic shales. The absence of Posidonia fossils places this unit on the upper levels of Posidonia shale formation. The smooth topography marks the spatial extent of the formation.

*4.4. Section C*

In section C, a great extent of the stratigraphy is also exposed. Eocene in age, thin- to medium-bedded, creamy to white calcareous mudstones, locally fossiliferous (nummulites) with lenticular chert intervals dominate the upper stratigraphic levels. The unit is extremely deformed and fractured as a result of the overturning of the whole northern fold limb of the Plataria syncline (Figure 9). Low-angle reverse faults are common. The lower stratigraphic units of the Eocene formation limestones are characterized by medium-bedded to massive limestones. Thinner beds are characterized by a mudstone texture, while thicker to massive beds present a wackestone to packstone texture. Cherts with nodular or irregular shapes are floating in the calcareous mass. Small-scale, syn-sedimentary folding suggests instability of the sediments at a primary depositional stage.

The Senonian formation's lithological character is quite similar to that of the other sections, and is characterized by thick-bedded to massive limestones (Figure 9B). A packstone and wackestone texture is dominant, while some thinner beds show a muddy texture. Sedimentary features such as planar or ripple cross-lamination are common, implying the mechanical deposition of current transported particles and indicates that these carbonates have not been precipitated in place. Fossil fragments are abundant, originating mainly from rudist species.

The upper stratigraphic levels of the Vigla limestones are dominated by thin- to medium-bedded, creamy to reddish mudstones (Figure 9F), with chert intervals identified as either layers or lenticular bodies. Gradually, the average thickness increases and alternations of thin- to medium-bedded mudstones with thick-bedded to massive wackestones to packstones are common. Three horizons of massive limestones were also identified in the specific stratigraphic level of the Vigla formation, with abundant fossil fragments (Figure 9E) and sedimentary structures, suggesting the involvement of gravity flow. The Vigla limestones overlay very thin- to thin-bedded, brownish-to-greenish, clayey shales alternating with brownish and greenish cherts. This unit corresponds to the Upper Shales unit of Vigla formation. The Vigla shales are on top of thin- to medium-bedded, creamy to light grayish mudstones. Intervals of layered cherts are common at the specific stratigraphic level. An upper Posidonia shale unit, though poorly exposed, was identified at the base of the aforementioned limestones, consisting of thin-bedded brownish shales with intervals of thin-bedded limestones. The gentle relief located eastwards of Vigla limestone exposures defines the spatial extent of the formation. The transition of Pantokrator limestones to Posidonia shale formation is characterized by medium-bedded limestones with intervals of lenticular cherts overlaid by thin-bedded shales alternating with thin- to medium-bedded brownish siltstones. (Figure 9C,D).

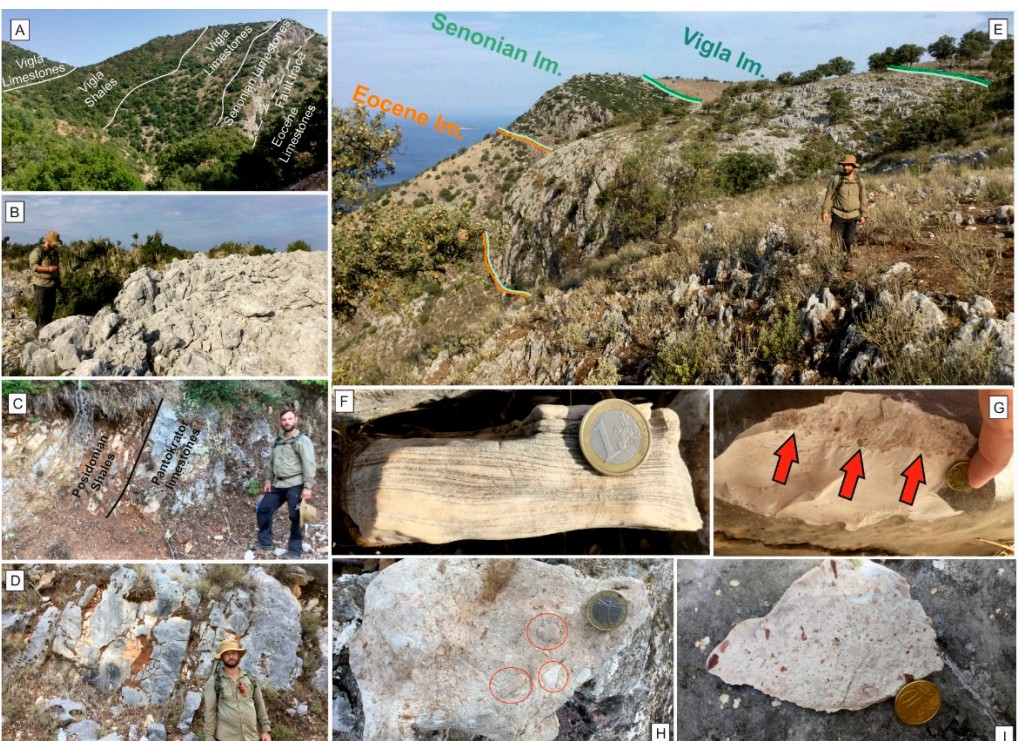

**Figure 9.** (**A**) General view of the exposed formations in the inverted stratigraphic sequence (older to the left, younger to the right). Notice the fault crossing the Senonian limestones. (**B**) View of the massive Senonian limestones. (**C**) Posidonia shale formation overlying the Pantokrator limestones. (**D**) The upper beds of Pantokrator limestones. Notice the whitish chert intervals. (**E**) General view presenting the lateral extent of the various lithostratigraphic units in the overturned flap. Colored lines represent the contact between different formations (**F**) Well-preserved ripple and planar lamination in Vigla limestone formation. (**G**) Image showing the erosive contact (red arrows) between a Senonian debrite and the underlying mudstone. (**H**) Outcrop calcidebrite sample from Vigla limestone formation. Notice the clasts floating in the muddy matrix, indicated by red circles. (**I**) Debrite hand specimen originating from the Senonian limestones.

### 4.5. Section D

In section D, the logged stratigraphy includes the transitional beds of the Eocene limestone to flysch deposits. Very thin- to thin-bedded marly shales, characterized by intense schistosity and deformation, are partially exposed, marking the transition from chemical to clastic sedimentation. Moving towards the lower stratigraphic levels, a sequence of thin- to medium-bedded, creamy to white calcareous mudstones, locally fossiliferous (nummulites) is present, similar to the other sections, corresponding to the Eocene limestones. Medium-bedded to massive limestones with intervals of thinner beds dominate the lower Eocene formation levels. A mudstone texture is typical for thin beds, while thicker to massive beds have been characterized as wackestones and packstones. Cherts with nodular or irregular shapes are floating in the calcareous mass.

The Senonian limestone formation is typically thick-bedded to massive (Figure 9H), classified as packstones to wackestones, while thin beds are mudstones. Sedimentary features such as planar or ripple cross-lamination suggest the involvement of turbidity currents in the depositional processes (Figure 9F). Rudist fossil fragments are also common in the stratigraphic intervals. The formation is susceptible to karstification, as is identified by the characteristic rough texture of the exposed surfaces.

The Vigla formation starts with alternations of thin- to medium-bedded limestones, characterized by a mudstone texture with medium- to thick-bedded wackestones to packstones. Thick-bedded horizons identified in the succession represent the lateral extent of

the same massive beds that are identified westwards in the specific stratigraphic levels (Figure 9I). Calcidebrites identified in the specific stratigraphic level are characterized by sub-angular to sub-rounded clasts floating in a calcareous mass. The size and the composition of the clasts varies. Biogenic components such as fragmented shells of rudists are common, while non-biogenic material is composed of clasts of limestone and chert (Figure 9G). At the lower stratigraphic levels, very thin- to thin-bedded brownish to greenish clayey shales, alternating with brownish and greenish cherts occur, resembling the Upper Vigla shales. Following the other measured sections, the Vigla shales overlay thin- to medium-bedded, creamy to light grayish calcareous mudstones. Chert intervals are also identified in the formation. At the lower stratigraphic levels, small-scale folding occurs. The base of the measured stratigraphy shares common characteristics with that of section C, with very thin- to thin-bedded, brownish to greenish clayey shales alternating with brownish silts overlapping the medium-bedded limestones.

*4.6. Section E*

Section E is located in the easternmost part of the study area. The Eocene limestones consist of medium-bedded to massive limestones. Thinner beds are characterized by a mudstone texture, while thicker to massive beds present a wackestone to packstone texture. Erosive surfaces and planar and ripple cross-lamination are indicative of a calciturbiditic origin. Cherts with a lenticular shape are also common.

The Senonian limestones, consistent with the lithological pattern of sections A2 to D, are characterized by thick-bedded to massive limestones. Packstone (Figure 9I) and wackestone texture is dominant, while some thinner beds are characterized by a muddy texture. Well-preserved erosive surfaces are found in thick beds (Figure 9G). A significant differentiation is marked in Vigla limestones, where thick-bedded calciturbidites have not been identified in the stratigraphic record.

At the specific section, Vigla formation consists of thin- to medium-bedded, creamy to reddish limestones characterized by a mudstone texture. Chert intervals are identified either as layers or as lenticular bodies.

**5. Aerial Imaging**

To access the lateral extent of the distinct lithostratigraphic units and to reveal the internal unconformities that developed during salt rise, a photogrammetric survey was performed using a DJI Phantom 4 + drone. The flight plans were designed and controlled using the Drone Deploy application. The survey resulted in the capturing of more than 3500 georeferenced images covering an extended area of 1.06 square kilometers (Figure 10). The resolution of the acquired images ranged from 1.8 to 2.7 cm/pixel.

For the processing of the data, Pix4D software 4.8 was used. The processing stage contained three subsections: (a) Initial Processing, which automatically extracts key points from the imported aerial images to compute the internal and external camera parameters using the software's advanced Automatic Aerial Triangulation (AAT) and Bundle Block Adjustment (BBA). A sparse 3D point cloud was computed and a low-resolution DSM and orthomosaic were generated and displayed in an initial quality control process. (b) The second stage of the analysis was to produce a dense 3D point cloud and a 3D textured mesh. (c) Finally, a Digital Surface Model (DSM) was produced accompanied by an orthomosaic image of the covered area. The results of the analysis and the interpretation from a geological point of view are presented in Figure 11.

According to the photogrammetric analysis, the Eocene and Senonian thick- to massively-bedded horizons can be traced at a distance of more than 1 km. Concerning the lithostratigraphic correlations of these horizons, the analysis indicated a normal type of contact with the underlying and overlying lithostratigraphic units. On the contrary, the Early Cretaceous Vigla limestones are characterized by internal angular unconformities of the thick- to massively-bedded horizons with the surrounding calcareous mudstones. Unconformity angles range from 15 to 25 degrees (Figure 4b). The lateral extent of the calcidebritic and

calciturbiditic beds is similar to that of the Senonian and Eocene formations. The presence of internal unconformities in Vigla limestones, together with the identification for the first time of calcidebrites/calciturbidites horizons in this formation, shows indications of platform steepening due to salt activity and possible combination with tectonic activity.

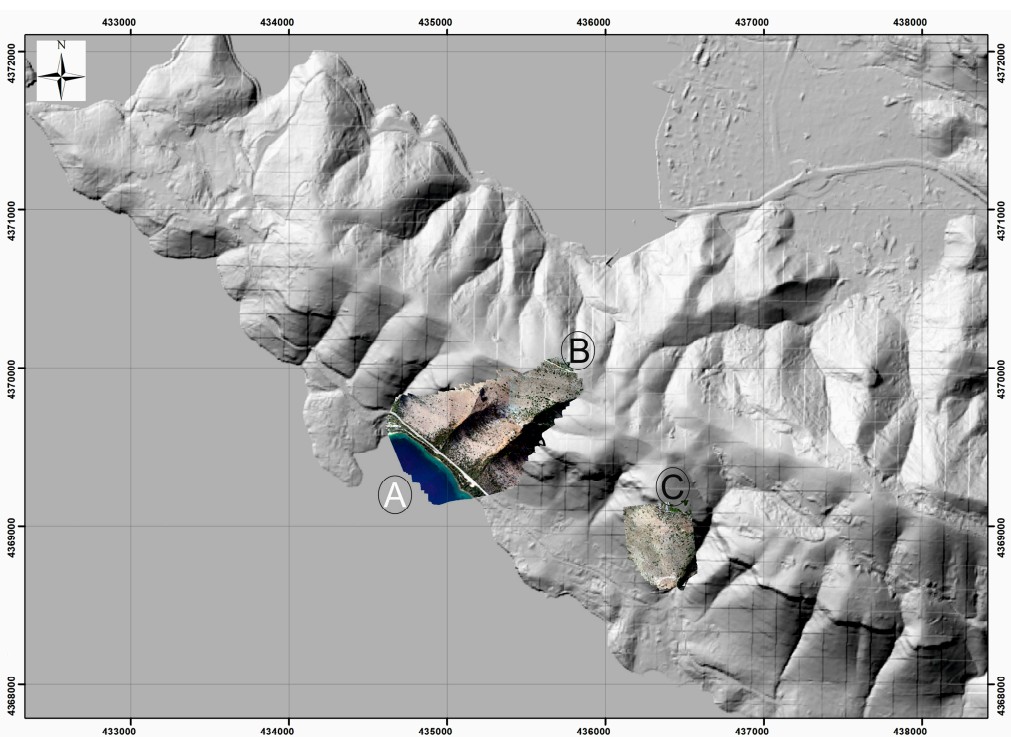

**Figure 10.** Locations of the produced models using UAV imaging (A, B and C represent the covered areas).

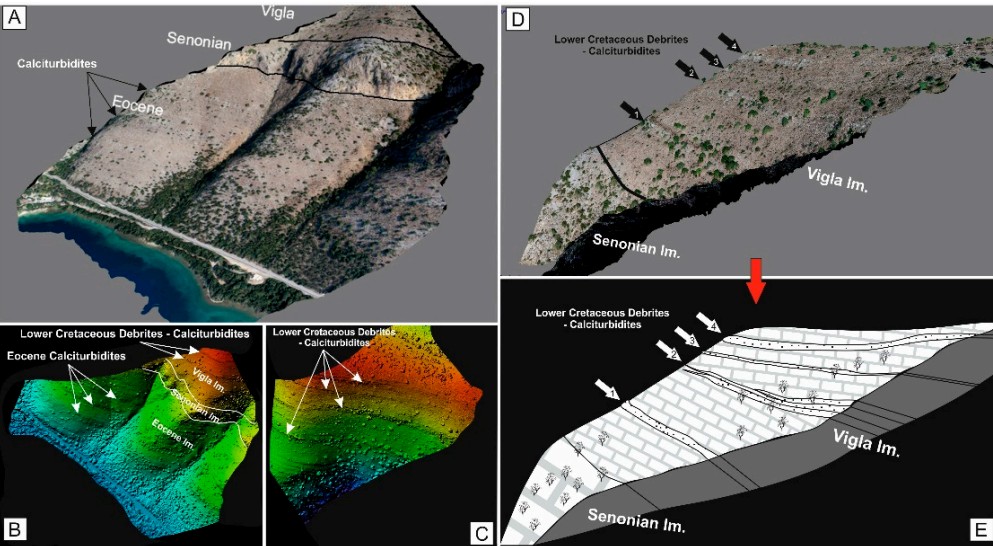

**Figure 11.** (**A**) Three-dimensional model showing the extent of Eocene calciturbiditic horizons. (**B**) DEM of the previous model showing how the calcidebrites/calciturbidites are recorded in the produced models. (**C**) Zoomed image of (**B**) showing the calcidebrites/calciturbidites of Vigla formation. (**D**,**E**) Model and graphical representation of Vigla calcidebrites/calciturbidites. Black and white numbered arrows indicate the different calciturbiditic/calcidebritic horizons.

## 6. Seismic Profiles

The structural-stratigraphic analysis was supported by a regional 2D depth-migrated seismic reflection profile, trending NE–SW, that was acquired in the vicinity of the study area. The IO17-12 seismic line, presented here, was acquired during the IO17 2D Seismic Campaign, which concluded in 2019. The IO17 Campaign was part of the contractual obligation for the first exploration phase for the Ioannina Concession under the Operatorship of Repsol Exploración Ioannina S.A. Greek Branch on behalf of the Repsol S.A. and Energean Oil & Gas joint venture. The acquired 2D seismic data were subsequently processed in Repsol's Subsurface Imaging center in Houston, U.S.A. (for details, see Appendix A). The data-driven interpretive processing sequence that resulted in the final 2D Kirchhoff Pre-Stack Depth Migrated Section, presented in Figure 12, was meticulously formulated, revised, and modified under the Repsol Exploration Department's supervision, incorporating all available information (well data, surface formation dip measurements along all IO17 seismic line tracks, etc.). Following the same approach, seismic interpretation was conducted honoring all available hard data (such as surface geological mapping and dip measurements, surface fault traces, well tops, etc.).

In summary, the processing objectives were as follows:

- Improve on the seismic image in the time domain (PSTM).
- Focus on the structural image in all areas of the sections.
- Provide consistency between datasets, from both legacy and new IO17 seismic acquisitions.
- Provide PSDM images and a pseudo-3D regional velocity model consistent with surface geology and well velocities.

The seismic profile depicts the regional structure up to a depth of 6 km. The upper part of the profile was seismically noisy and not much information was provided from that; thus, more surface structural data were needed to complement and complete the seismic profile structure. The complete Triassic to Eocene carbonate sequence was affected by the compressional deformation, developing an imbricate thrust system, consisting of thrusts that route and merge on a low-angle decollement horizon, represented by the evaporites. Along the eastern part of the decollement horizon, salt appears to be sheared and a salt weld marks the detachment from its source layer. This can provide some evidence that below that structural level in the basement there must be a thicker feeder evaporitic unit, from which the intrusive diapiric structure developed along the thrust surface. Alternatively, the basement can be a thrust-duplex structure of the Ionian unit or carbonates of the Pre-Apulian unit [102]. To the west, the fore-thrust brings the evaporites and Triassic breccia formation over a recumbent syncline fold that forms part of the overturned forelimb of a larger-scale, asymmetric to the west, box-type fold that is cored by the intrusive evaporites and Triassic breccia. A similar structural style can be seen further north in the exposed Dumre diapir in Albania, where diapirism is linked to a regional overthrusting to the west, truncating an overturned syncline consisting of Jurassic-to-Paleogene strata below the western evaporite thrust margin [103,104].

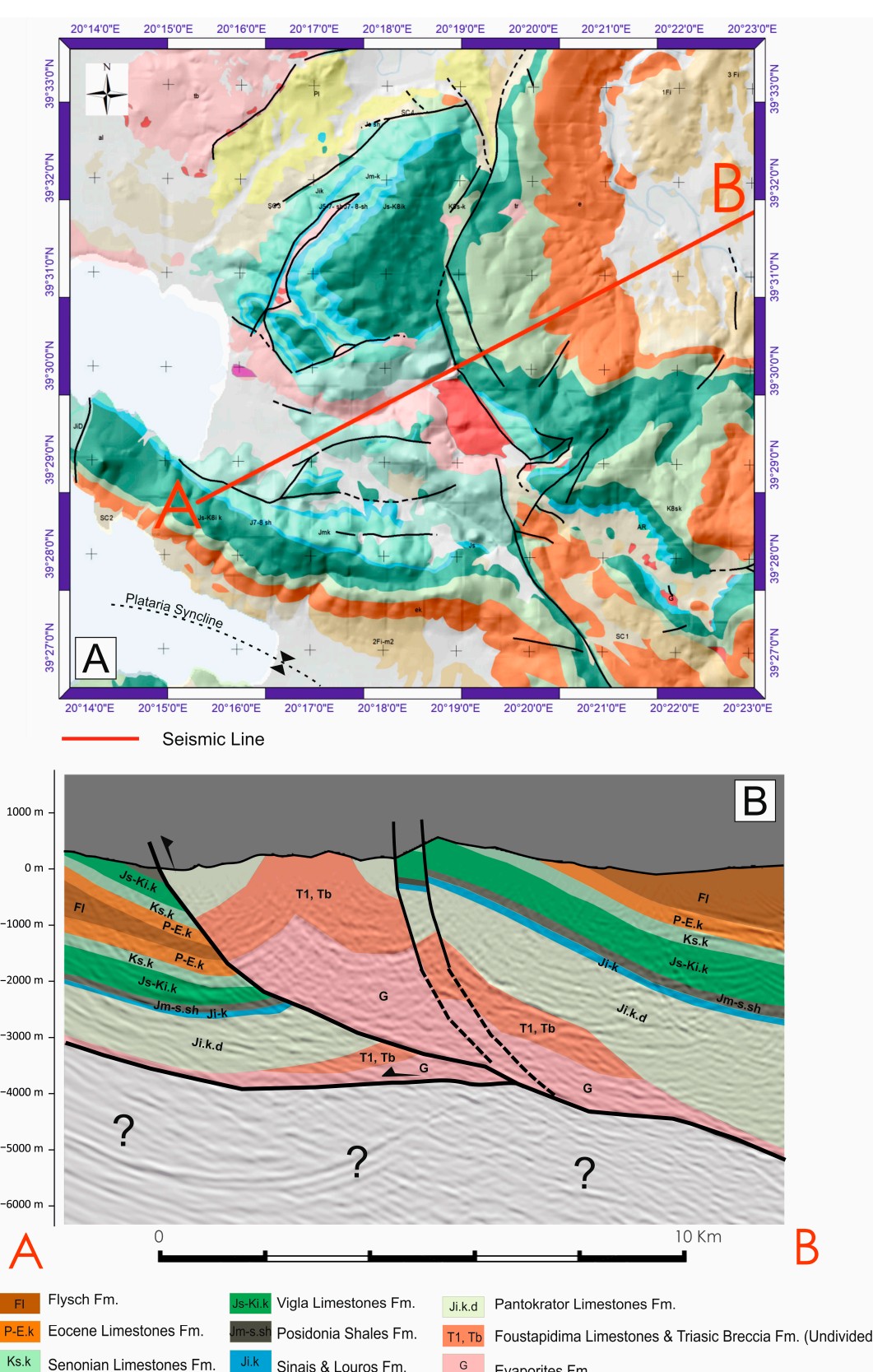

**Figure 12.** (**A**) Seismic line extent in the study area (red line). (**B**) Interpretation overlaid on the IO17-12 seismic line (?: Unknown rock type).

## 7. Depositional and Structural Evolutionary Model

A depositional and structural evolution model of the study area, based on the sedimentological, geophysical and remote sensing data previously presented, as well as on previous studies in the Ionian zone [69,72,82,83], is summarized in the conceptual model of Figure 13.

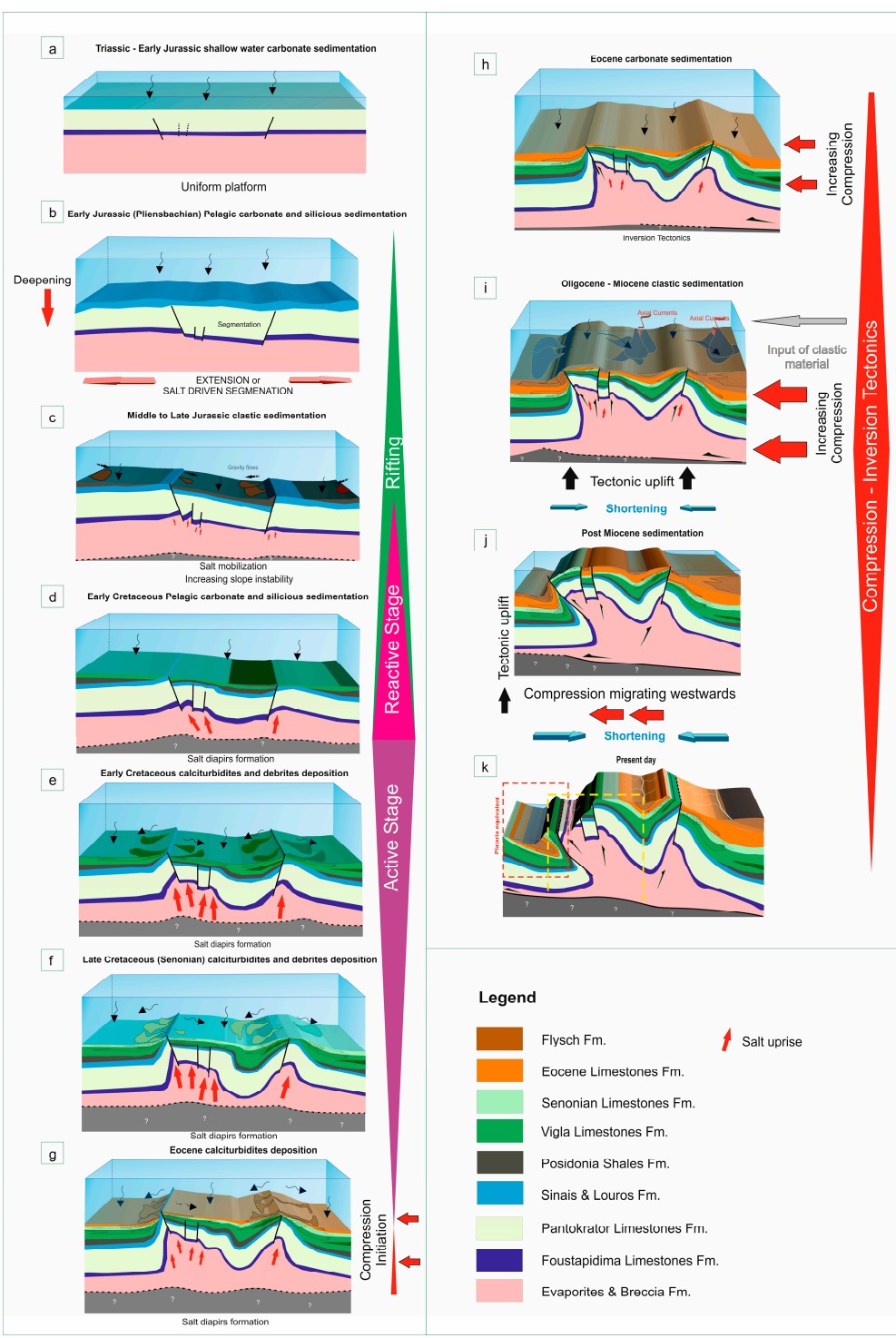

**Figure 13.** Structural-depositional conceptual evolutionary model of the study area. The Plataria overturned syncline and the seismic profile equivalents are shown in block diagram (**k**) (red and yellow dashed-line rectangles, respectively). White question marks indicate unknown rock type below evaporites.

During the Triassic–Early Jurassic, a shallow water, rifted carbonate platform (Figure 13a) dominated the area, represented in the stratigraphic record by Pantokrator limestones. The platform overlay a thick, flat-layered and almost isopachous evaporitic substratum of Early to Middle Triassic age. Poorly exposed Foustapidima limestones are considered as an intermediate unit between the evaporites and the shallow water carbonates [105–107]. The tectonic regime continued gradually to an extensional phase (Figure 13b), which has been considered as Pliensbachian in age (syn-rift phase) [71,108]. Although these traditional interpretations placed the initiation of the syn-rift phase at this time, alternative interpretations support the existence of a Triassic rifting, based on field and seismic observations from NW Epirus, and lack of evidence for major faulting affecting the early Jurassic successions [82,83] (Figure 14). Extension caused the gradual deepening of the basin, favoring the deposition of thin to medium-bedded pelagic limestones with chert intervals (Siniais and Louros Limestones). The segmentation of the platform in various blocks of a wedge-shaped geometry (Figure 13b) controlled the deposition of various lithological types (pelagic limestones, Ammonitico rosso beds, Posidonia shales, limestones with filaments), reflecting the heterogeneity of the depositional conditions across the basin [69,72]. Additionally, the fault-controlled basin subsidence caused stretching and expelled the underlying salt by differential loading.

In the Plataria area, Posidonia shales are the dominant lithologic type of the specific evolutionary stage. The accumulated sediments of phases 1 to 3 reached a significant thickness (see Figure 3: Foustapidima limestones 50–150 m, Pantokrator limestones >1000 m and syn-rift facies 20–200 m) for density inversion to occur, and are ready to trigger salt uprise by buoyancy forces when external forces are applied. Previous studies suggest that to initiate salt rise at least 650 m of sediment thickness is required for an overlying uncompacted carbonate platform, while for siliciclastics the required thickness is 1500 m [2,109,110]. The primary factor that influences the formation and evolution of salt structures is the differential loading, as suggested by [111–113], as well as the pattern of the sub-salt basement structure (i.e., uplift and slope flanks). At this stage (Figure 13c), a reactive mode of diapirism [19] can be implied, regarding that a buried salt layer formed a diapir in reaction to extension, favored by the aforementioned density contrast and/or differential loading.

During extensional deformation, the overburden is lengthened and thinned, allowing salt to rise and fill the space below the tilted footwall fault blocks and later rise below the graben axis. The effect of salt diapirism is not yet clearly recorded in the sediment types (Figure 13d), where lower Cretaceous pelagic limestones (Vigla Formation) are deposited. At the deeper parts of the segmented sub-basins and on the tilted hanging-wall blocks, the Vigla shales were deposited. This tilting and the differentiation in the sedimentation pattern imply that diapiric intrusions (i.e., horn-type below the footwall blocks), which are activated by the extensional deformation of the syn-rift phase (reactive stage), start to play a key role in the sedimentation processes (Figure 14).

Extensional deformation during the syn-rift stage [69,72] has resulted in the weakening of roof strength, forming outer-arc bending on top of the diapiric structures, while the wedge-shaped geometry introduced anisotropic conditions. Additionally, the accumulation of sediments in the basin differentially increased the average density and thickness of the overburden. Evaporitic intrusions started to pierce the overburden, bending the roof and affecting the depositional patterns. At this stage, an active diapiric phase of halokinesis [2] was established in the area, controlled by sediment accumulation rates and the degree of anisotropic conditions of the roof [114]. Intervals of thick- to massive-bedded limestones started to form in the stratigraphic record, characterized by sedimentary features (e.g., erosional surfaces, planar and ripple cross-lamination), which indicates deposition by turbidity currents. Syn-sedimentary slump events also occurred, suggesting a depositional environment of unstable conditions originating mainly from the salt rise (Figure 13e).

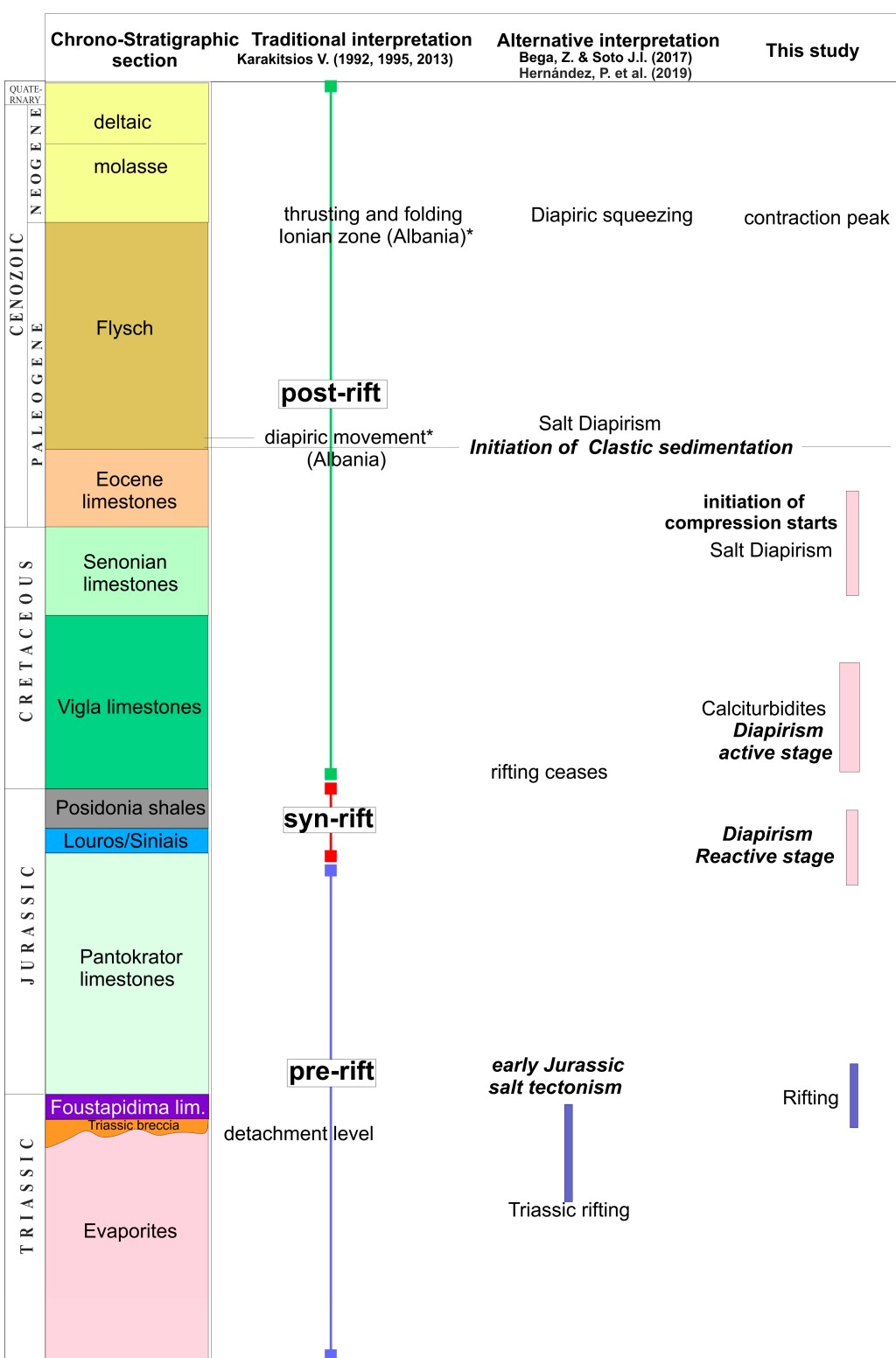

**Figure 14.** Simplified chronostratigraphic correlation of available interpretations for the Ionian zone, regarding the tectono-stratigraphic and salt tectonics processes, suggested by various researchers and this study. Data derived from key references are listed in the figure [69,71,72,82,83] (* represents diapiric movement in Albania region).

The effect of the salt uprise in the study area is imprinted in the whole Cretaceous carbonate succession with intervals of calciturbidites and calcidebrites (Figure 13f). Internal unconformities and onlap structures, observed in the studied sections, suggest the folding of the roof due to a halokinetic uprise, as is also evidenced in the Vigla section by the detailed 3D photogrammetric model (Figure 11).

This fact, supported also by the available seismic profile, the surface cross sections and structural measurements, enabled us to identify a basal megaflap [6,115], consisting mainly of Late Triassic carbonates and Jurassic shales, in the studied area. Salt uprise is inferred to have slowed down and ceased gradually during the Early Eocene (Figure 13g), where the final occurrences of the fine-grained calciturbidites were found in the stratigraphic record (Figure 14). Monotonous, pelagic thin- to medium-bedded mudstones overlie the aforementioned calciturbidites, followed by the deposition of the Oligocene flysch sediments. The active diapiric phase ceased probably due to the increase in the overburden thickness that exceeded the threshold thickness for salt mobilization [2] or a reduction in the triggering external tectonic forces.

As salt rise ceased, contractional deformation started to affect the basin configuration, introducing a phase of tectonic inversion, where the previously extensional structures that favored reactive and active diapiric phases were then reactivated as thrust faults. Uprising salts enhanced block movements across the pre-existing fault surfaces, acting as local décollement horizons (Figure 13h,i). A typical fold-and-thrust belt regime was established in the area, developing asymmetric synclinal and salt-cored anticlinal structures. The salt intrusion of the previous stages had already formed upturned flaps flanking the diapiric body to an almost sub-vertical inclination of the strata. Compressional deformation increased further from the dips, resulting in the complete overturning of fold limbs (Figure 13j,k). The combination of salt-driven inversion tectonics provided favorable conditions for the formation of structural traps.

## 8. Conclusions

A Triassic platform was compartmentalized by a combination of extensional and salt tectonics. Platform block segments were intruded by salt diapirs during a regional extensional phase, resulting in the control of sedimentation partially by gravity flows in a marginal-to-slope setting. Internal unconformities suggest that diapirism started to control sedimentation processes from the Lower Cretaceous to Early Eocene. Presence of calciturbidites, and to a lesser degree calcidebrites, indicate the instability of the platform caused by diapir rise, triggering gravity flows. From the Late Eocene to Miocene, compressional deformation and establishment of a foreland fold-and-thrust belt in the external Hellenides started to control sedimentation. The carbonate sedimentation ceased and clastics deriving from the emerging fold-and-thrust belt started to deposit.

The almost-vertical limbs of the rim synclines that formed adjacent to diapirs were overturned due to compression, resulting in a complete inversion of the stratigraphic sequence. A basal megaflap configuration can be implied based on the seismic profiles and outcrop structural data. Inversion tectonics have overprinted salt tectonics effects to a significant degree. Syn-sedimentary deformation resulted in the accumulation of debritic and turbiditic layers, which have a sedimentary texture that could potentially affect reservoir quality at the specific stratigraphic levels. Post-depositional diagenetic processes played a key role in the final product; however, studies in similar settings support the concept that gravity flow deposits can have an increased potential of primary porosity preservation. The compressional regime established in the area from Late Cretaceous favored fracture porosity, potentially increasing the reservoir quality of the studied carbonate sequence.

**Author Contributions:** Conceptualization: I.V., S.K., P.K., P.H.-J. and R.P.-M.; methodology: I.V., P.K., P.H.-J., R.P.-M., I.K. and H.T.; seismic data interpretation: C.T., P.H.-J., R.P.-M., J.P.P.-G., I.V. and S.K.; data validation: I.V., C.T., I.K., H.T., P.H.-J., R.P.-M. and J.P.P.-G.; data curation: I.V., S.K. and C.T.; writing—original draft preparation: I.V. and S.K.; writing—review and editing. All authors have read and agreed to the published version of the manuscript.

**Funding:** This research received no external funding.

**Institutional Review Board Statement:** Not applicable.

**Data Availability Statement:** All data are available and included in this article.

**Acknowledgments:** The authors wish to acknowledge Repsol S.A. and Energean Oil & Gas for the permission to use and present part of the data acquired in the context of geological studies of the First Exploration Phase for the Ioannina Concession under the Operatorship of Repsol Exploración Ioannina S.A. Greek Branch. Additionally, the authors thank the members of the G&G departments of both companies for helpful discussions and suggestions for the improvement of the current work.

**Conflicts of Interest:** The authors declare no conflict of interest.

## Appendix A. Seismic Data Processing

The seismic data time processing was performed in two separate phases: Phase I included more than 1100 km of legacy seismic data of various campaigns and various acquisition parameters such as Vibroseis or Dynamite source, with relatively low active folds (24–60) and maximum offsets (up to 5.5 km). Phase II, on the other hand, concentrated on the newly acquired 400 km of seismic data of the IO17 Campaign, with much improved acquisition parameters such as a higher load/depth Dynamite source, greater active fold (167 CMP/1002 Channels) and a maximum offset of 10 km.

For Phase I—legacy lines, processing started from the raw datasets with geometry and underwent coherent and anomalous noise attenuation. Using an in-house structural smoothing algorithm, the 2D shots from the acquisition were sorted in a way that the shots resembled a 3D volume in space (shot stations becoming inline numbers and receiver stations becoming crossline numbers). This further enhanced the coherency of deeper events and allowed for better structural model building. For Phase II, the processing also started with raw data for the initial model building, and later, in more advanced stages, were replaced with the preprocessed dataset from the Repsol/CGG Madrid processing center. This dataset underwent the same pseudo 3D structural smoothing sequence as the Phase I datasets.

As far as the seismic data depth processing was concerned, velocity model building was performed by using commercial software suites, along with Repsol's proprietary algorithms. A data-driven workflow for migration velocity estimation was implemented, starting with near surface modeling (NSM) using first break arrival times, including long offsets. Diving wave tomography provided a model that was consistent with surface geology and available well velocities. Flooding models were created based on the deepest NSM velocities as permitted by the maximum depth of penetration, which allowed us to image the top of carbonate sections. Images corresponding to the flood models were used to provide interpretation for guiding deeper model building in an iterative workflow using a structural tensor algorithm which incorporated the measured structural dips at the surface. Such dips acquired as part of the acquisition campaign were instrumental for controlling the interpretation and, consequently, the velocity model. The final velocity model allowed us to improve the seismic imaging around several structures developed by salt tectonics that are outcropping in the area.

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
