# Peer review of "Implications of Salt Diapirism in Syn-Depositional Architecture of a Carbonate Margin-to-Edge Transition: An Example from Plataria Syncline, Ionian Zone, NW Greece"

_applsci, doi:10.3390/app13127043_

Round 1

Reviewer 1 Report

To the editor of

Applied Science

Manuscript Number: applsci-2424282

Title: Implications of salt diapirism in syn-depositional architecture of a carbonate margin-to-edge transition: an example from Plataria syncline, Ionian zone, NW-Greece.

Article Type: Original Article

Authors: Ioannis Vakalas , Sotirios Kokkalas, Panagiotis Konstantopoulos, Constantinos Tzimeas, Isidoros Kampolis, Helen Tsiglifi, Ruben Perez, Pablo Hernandez, Juan Pablo Pita-Gutierrez

Reviewer score: minor revisions

Respected Editor,

Dear Authors,

I reviewed the manuscript by Vakalas and co-authors entitled “Implications of salt diapirism in syn-depositional architecture of a carbonate margin-to-edge transition: an example from Plataria syncline, Ionian zone, NW-Greece”.

The manuscript illustrates a detailed study of an overturned syncline that takes advantage of both field, remote (UAV) and subsurface data. Thanks to these data, the effects of salt tectonics on both sedimentary and tectonic processes are discussed and a possible evolutionary model of the study area is proposed.

The topic of the paper is of wide interest and suitable for Applied Science readership because it starts from a detailed study on a limited/local structure, but it allows to derive general observations and evolutionary reconstructions that can be methodologically applied in a number of different study areas.

I found the paper well organized and written, the integration between different kind of data is clear and the discussion and conclusions are supported by observations and data analysis.

In my opinion some minor revisions can be carried out by the Authors to improve the paper and its readability. I refer in particular to some aspects of the pictures and to minor changes in some of them. Even if I’m not a mother tongue I found the English clear; in some places I suggested to rephrase or to change some words.

All these suggestions are contained in the annotated PDF associated with this letter.

For all the previous reasons I think that the manuscript is of interest for Applied Science readership and can be accepted for publication after minor revisions.

Thank you for giving me the opportunity of reviewing this paper.

Best regards

Giovanni Toscani

University of Pavia, ITALY

very minor points, mainly typos, have been indicated in the annotated pdf

Author Response

Responses to reviewers’ comments

Article: Implications of salt diapirism in syn-depositional architecture of a carbonate margin-to-edge-transition: an example from Plataria syncline, Ionian zone, NW-Greece.

By Vakalas, I., Kokkalas, S., et al. 2023

Reviewer 1

Comments:

Even if coordinates are present, I would add a small inset or a small picture of the entire Greece, just for non-EU readers. May be a small tectonic sketch would be of help.

Why not adding a structural sketch of the extended study area and a simplified representative cross section taken form literature to help the readers unfamiliar with these structures to have a better view of them?

Response to main comments by the reviewer.

  • Figure 1 is updated with an inset of the entire Greece, for non-EU readers. Additionally, a simplified structural section has been added.

  • Figure 2 is updated with a scale bar inside the map frame and the missing black rectangle pointing to the north limb of the study area.

  • Fault kinematics can only be implied in map frame extend, so a more detailed classification of fault types has been avoided.

Comment: a sketch of these processes would be of help and would make the paper even more cited thanks to this initial brief "overview". Probably you can find something useful looking at the papers of Oriol Ferrer or others (Vandeville, Mark Jackson) that worked a lot on salt tectonics using simplified analogue models. Just a suggestion...

Response: We thank the reviewer for his suggestion, but since these mechanisms are widely referred in salt tectonics books and the literature, we think that would be like a repetition to reproduce the same classic models just for reference. If we didn’t have so many figures in the ms, we might consider it.

The rest of the comments regarding syntax, typo and rephrasing text, have been included in the revised version. The comments and suggestions of the reviewer are welcomed and his review is greatly appreciated.

Reviewer 2 Report

This is an important paper that provides through the use of drone photogrammetry and 2D processed seismic new insights into sedimentation and salt tectonic history of the Ionian Zone of western Greece. I would expect it to become a well-cited study. But this version of the paper reads like a first draft, with inconsistencies between sections perhaps written by different co-authors. I think almost all the science is sound, which is why I have chosen “minor revision” but the presentation will need a lot of work that could be classified as “major revision”.

There are three weak parts of this paper that will require substantial revision including modifications to figures.

1. Inconsistent stratigraphic nomenclature between figures 2, 3, 12 and the text.

2. The description of the field sections is confusing with regard to what is the stratigraphic succession present and which parts of it are structurally inverted. Some repetition might be removed by reference to previously described field sections.

3. The evolutionary model is not well argued nor referenced to the literature (please be selective, cite only the necessary key literature here), and reasonable alternative hypotheses are not considered or arguments given for their rejection.

I thought that the number of references in sections 1 and 2 was excessive (almost 100) and that it provided the reader with little guidance on what were the key pieces of previous work. I agree that it avoids offending colleagues to provide so many references.

It is not possible with the resolution of Fig. 11 to confirm the authors’ geological interpretations of the photogrammetry survey. These data should be submitted as Supplementary Material. I can see no reason why they should be commercial confidential.

DETAILED COMMENTS KEYED TO LINE NUMBERS

33          no ‘s for inanimate objects. Write “enhanced the fracture porosity of the carbonates”. Also “could eventually affect the”

42-43    what reduction?   You mean “the need for reduction …”  It has not happened yet!    “the possibility of geological storage of carbon dioxide”

49          “various salt-tectonic structures”   The previous sentence deals with sedimentary rocks, so that structures might well apply to sedimentary structures.

79          terming it a “key” area seems to require a brief statement of why it is a “key” area.

95          whole extent of

104        respectively

105        were deposited

108        “developed” is the wrong word. Original deposition is part of the development. Perhaps “emplaced”.    Also, Cenozoic is preferred to Tertiary (International Commission on Stratigraphy)

Fig. 2     Alluvium, not alluvials       This has a “lazy scale” requiring the reader to work hard at it. Show 2 km not 2¼ km.  Posidonia not Posidonian (as the text uses Posidonia) (but later the text uses Posidonian – please be consistent).

125        limb

125        I could not find the (black rectangle). Please make its position clearer.

140        “phases” should be deleted

140-142              These new ideas make much more sense to me than the “pre-rift”, “syn-rift” and “post-rift” in Figure 3 and subsequent text. Do the authors accept these new ideas? As noted below, these new ideas should be included in the Discussion and proposed new Fig. 14, where arguments against this interpretation should be outlined.

Fig. 3.    Cenozoic not Tertiary. Ideally use the same terminology as in Fig. 2, where for example the Foustapidima Limestone is included in the Pantokrator Limestones Fm.

158        were still active (past tense)

160-182              most of this needs to be in the past tense, it is describing events at some time in the past. If there is a description of a rock, then that is in the present tense.

163        During Eocene time, clastic material supply was reduced …

167        was followed by a gradual passage to …

174        here “divide” (present tense) is correct, because it is a description of the rocks

185        only a person can propose.  So either  “… data led [72], [98] to propose a moderate …”  OR “data were interpreted to indicate a moderate…”

200        Figure number out of sequence. I suggest you do not reference the seismic figure here. MDPI editorial staff are unforgiving!

203        Either add a bar scale or note in caption “UTM grid spacing 1 km.”

211        I suggest to add: “Sections are described from topographic top to base.” It is normal convention to describe field sedimentary rocks oldest first, and the reader may be momentarily confused by your approach.

218        Clarify: does “succeeded” mean underlain or overlain by …

222        This seems seriously muddled. How can Senonian limestones formation be deposited on top of the previous unit, which “succeeded” Eocene deposits. The Senonian limestones may have been tectonically emplaced on top.

224        Fossil fragments including rudists are common

230        turbiditic

233-236              Are these onlaps visible in the field? How do you recognise them? Is it from your photogrammetry, or is this a seismic interpretation? Whichever, it needs a little more explanation and may be out of place here.

240        I am not questioning that some chert layers are useful indicators of soft-sediment deformation (e.g. image F). But I cannot see any evidence of such deformation in image D. Either indicate the deformation or modify the caption. Such discontinuous beds of chert are not necessarily evidence of boudinage.

244        This becomes more confusing as it is (well) described in conventional order from base to top.

250        You see a slump, you infer an event. Better in this context to write “Syn-sedimentary slumps suggest …”

271        Part of my confusion is because “succeeded by” has a time connotation. Tsipras was succeeded by Mitsotakis. The best spatial term is “overlain by”

276        probably

290-295              This seems to depend on the seismic, which has not yet been presented. Probably more appropriate in a later Discussion section.

296-297              I am having a hard time linking this to the stratigraphy shown in Figs. 2 and 3. Try to use a consistent terminology in the figures and text of this paper.

357        does “following” mean overlying? Or underlying? Or that it is similar to the other sections.

373        rudists

433-440              Are the onlapping beds described in lines 233-236 based on photogrammetry?

Fig. 12   The stratigraphic units in panel (B) do not match Fig. 2, nor Fig. 3, nor the text on stratigraphy. Furthermore, the lettering in this figure is too small, requiring at least a 200% magnification to read.

471        was seismically noisy and not much

474        why ‘        around decollement  ?

480        spell out fm.  as elsewhere in paper

481        recumbent

491        represented in

495        What is the evidence that the phase is extensional. Why is the observed facies deepening not the result of salt withdrawal ?    [OK, this argument is dealt with in lines 502-510, but perhaps some rewriting would help make the point earlier].

496-501              I don’t recall this facies segmentation being presented earlier. These statements need selected reference to the literature.

516        ‘fill the space’   What space?

517-518              Why are the highs with limestone not the result of salt tectonics?

526-530              Is there any evidence for the localisation of the carbonate debrite facies around salt diapirs? What about the hypothesis that it is deepening of the basin due to salt withdrawal that allows steep platform margins to develop, strengthened and steepened by reefs, and it is this combination of subsidence and biological changes in reef architecture rather than salt rise that results in turbidites and mass transport deposits? This is not my field, but I recall papers about changes in the style of Jurassic continental margin carbonates going from gentle ramp to steep reef through time. ? Leinfelder, R. R., D. U. Schmid, M. Nose, and W. Werner, 2002, Jurassic reef patterns—The expression of a changing globe, in Phanerozoic Reef Patterns: SEPM, p. 465–520, doi:10.2110/pec.02.72.0465.    Pierre, A., C. Durlet, P. Razin, and E. H. Chellai, 2010, Spatial and temporal distribution of ooids along a Jurassic carbonate ramp: Amellago outcrop transect, High-Atlas, Morocco: Geological Society, London, Special Publications, v. 329, no. 1, p. 65–88, doi:10.1144/SP329.4.

532-533              remind the reader of your specific evidence for onlap (see comment on lines 233-236) and reference Fig. 11 for unconformities.

Fig. 13. A speculative base salt horizon would be useful in panels 1-7

It would make the arguments easier to follow if there were an additional figure 14. It would show the stratigraphic column and nomenclature, the classic pre- syn- and post-rift divisions; the proposed divisions of [82] to [84] with timing of salt movement; and your proposed model with timing of extensional periods and of salt uplift and piercing.

565-568              past tense, as in the beginning of this paragraph

571        by convention, no references in Conclusions. It should have been sufficiently discussed in the Discussion. Some of these good words might be moved to the Discussion.

586        Why is the not applicable. Indicate that most data are commercial confidential and get permission to show the photogrammetry results at a usable scale.

641        None of the references is in the (horrible) MDPI house standard, with the year between the italicized journal abbreviation and the volume number.

665        use symbol functions in Word to present these names correctly (or copy characters from line 689)

David J. W. Piper

[email protected]

see suggestions in my report above

Author Response

PLEASE SEE ALSO ATTACHMENT

Responses to reviewers’ comments

Article: Implications of salt diapirism in syn-depositional architecture of a carbonate margin-to-edge-transition: an example from Plataria syncline, Ionian zone, NW-Greece.

By Vakalas, I., Kokkalas, S., et al. 2023

Reviewer 2

Comment: This is an important paper that provides through the use of drone photogrammetry and 2D processed seismic new insights into sedimentation and salt tectonic history of the Ionian Zone of western Greece. I would expect it to become a well-cited study. But this version of the paper reads like a first draft, with inconsistencies between sections perhaps written by different co-authors. I think almost all the science is sound, which is why I have chosen “minor revision” but the presentation will need a lot of work that could be classified as “major revision”.

Response: Firstly, we would like to thank the reviewer for the constructive and thorough review of our manuscript. In the revised version we adopted almost all of his suggestions to address adequately these inconsistencies of the first version.

Comment: There are three weak parts of this paper that will require substantial revision including modifications to figures.

  1. Inconsistent stratigraphic nomenclature between figures 2, 3, 12 and the text.

Response: All inconsistencies between figures and text were revised and updated figures were added in the revised version (Figs 2, 3, 12).

In Figure 2, Foustapidima Limestones are not exposed in the surface.

Generally, Foustapidima limestones are very poorly exposed, and when found usually this formation consists of blocks into the Triassic breccia. For this reason, Foustapidima Limestones in Figure 12 is merged into a single symbol with the Triassic Breccia.  

In Figure 3, Triassic breccia fm. has been added.

In Figure 12 Triassic Breccia (Tb) Foustapidima limestones (T1) and Evaporites (G) were corrected in order to follow the legend.

  1. The description of the field sections is confusing with regard to what is the stratigraphic succession present and which parts of it are structurally inverted. Some repetition might be removed by reference to previously described field sections.

Response: We agree with the reviewer. In the revised version we fixed that confusion, clarifying that we study an overturned inverted stratigraphy, where stratigraphically older units are now seen on field over the younger ones. Despite that for consistency, all field sections are now described in the same way from the stratigraphically top (Eocene) to the bottom (Early Jurassic-L.Cretaceous).

  1. The evolutionary model is not well argued nor referenced to the literature (please be selective, cite only the necessary key literature here), and reasonable alternative hypotheses are not considered or arguments given for their rejection.

Response: Additional arguments explaining the stages of the model were added. The previously established alternative hypothesis has been discussed in the discussion and a new figure (Fig. 14) is added in the revised version with key references and main findings (in comparison to other models) regarding the evolutionary stages of the Ionian zone in terms of tectonics, salt activity and key lithostratigraphic data.

Comment: It is not possible with the resolution of Fig. 11 to confirm the authors’ geological interpretations of the photogrammetry survey. These data should be submitted as Supplementary Material.

Response: A higher resolution image of Fig. 11 is added in order to give a better insight on the presence and extend of the calciturbidite horizons and local unconformities discussed in the text.

DETAILED COMMENTS KEYED TO LINE NUMBERS

All reviewers’ minor comments from lines 33-140 (grammatical, typo, rephrasing) have been applied. Tertiary was changed to Cenozoic (as is pointed out by the reviewer and ICS)

Fig. 2     Alluvium, not alluvials. This has a “lazy scale” requiring the reader to work hard at it. Show 2 km not 2¼ km.  Posidonia not Posidonian (as the text uses Posidonia) (but later the text uses Posidonian – please be consistent).

Response: All changes were made in Figure 2 and text. We used consistent name for formations in figures and text.

125        I could not find the (black rectangle). Please make its position clearer.

Response: Black rectangle has been added.

140-142              These new ideas make much more sense to me than the “pre-rift”, “syn-rift” and “post-rift” in Figure 3 and subsequent text. Do the authors accept these new ideas? As noted below, these new ideas should be included in the Discussion and proposed new Fig. 14, where arguments against this interpretation should be outlined.

Response: As stated above in major comment point 3, the established alternative hypothesis has been discussed in the discussion and a new figure (Fig. 14) is added in the revised version with key references and main finding regarding the evolutionary stages of the Ionian zone in terms of tectonics, salt activity and key lithostratigraphic data.

200        Figure number out of sequence. I suggest you do not reference the seismic figure here. MDPI editorial staff are unforgiving!

Response: Done, we referred to section for further details than the out of order figure.

203        Either add a bar scale or note in caption “UTM grid spacing 1 km.”. Response: Done

211        I suggest to add: “Sections are described from topographic top to base.” It is normal convention to describe field sedimentary rocks oldest first, and the reader may be momentarily confused by your approach.

Response: Similarly, as was mentioned above, in the revised version we fixed that confusion, clarifying that we study an overturned inverted stratigraphy, where stratigraphically older units are now seen on field over the younger ones. Despite that and for consistency, all field sections are now described in the same way from the stratigraphically top (Eocene) to the bottom (Early Jurassic-L. Cretaceous).

222        This seems seriously muddled. How can Senonian limestones formation be deposited on top of the previous unit, which “succeeded” Eocene deposits. The Senonian limestones may have been tectonically emplaced on top.

Response: This confusion was made due to the wrong use of word ‘deposited’. Senonian is found above younger units due to the inverted stratigraphy in the overturned Plataria syncline. Section descriptions are now done homogeneously from the stratigraphic top to bottom. In the revised version all these issues are clarified in the text and figure captions.

233-236    Are these onlaps visible in the field? How do you recognise them? Is it from your photogrammetry, or is this a seismic interpretation? Whichever, it needs a little more explanation and may be out of place here.

Response: Yes, these onlaps are visible in the outcrop scale and in the photogrammetry study. We analyzed that a bit more now in the aerial image section and in the revised figure 11.

240        I am not questioning that some chert layers are useful indicators of soft-sediment deformation (e.g. image F). But I cannot see any evidence of such deformation in image D. Either indicate the deformation or modify the caption. Such discontinuous beds of chert are not necessarily evidence of boudinage.

Response: We added a new image on D and provide higher resolution of the figure 6

290-295              This seems to depend on the seismic, which has not yet been presented. Probably more appropriate in a later Discussion section.

Response: We agree and we moved the text to the discussion chapter.

Fig. 12   The stratigraphic units in panel (B) do not match Fig. 2, nor Fig. 3, nor the text on stratigraphy. Furthermore, the lettering in this figure is too small, requiring at least a 200% magnification to read.

Response: All these were done in the revised section. See earlier response to similar comment above.

496-501              I don’t recall this facies segmentation being presented earlier. These statements need selected reference to the literature.

Response: We used selected reference for that part.

516     ‘fill the space’   What space?

Response: We clarified this in the revised section with more detail. (see revised version)

526-530              Is there any evidence for the localisation of the carbonate debrite facies around salt diapirs? What about the hypothesis that it is deepening of the basin due to salt withdrawal that allows steep platform margins to develop, strengthened and steepened by reefs, and it is this combination of subsidence and biological changes in reef architecture rather than salt rise that results in turbidites and mass transport deposits? This is not my field, but I recall papers about changes in the style of Jurassic continental margin carbonates going from gentle ramp to steep reef through time. ? Leinfelder, R. R., D. U. Schmid, M. Nose, and W. Werner, 2002, Jurassic reef patterns—The expression of a changing globe, in Phanerozoic Reef Patterns: SEPM, p. 465–520, doi:10.2110/pec.02.72.0465.    Pierre, A., C. Durlet, P. Razin, and E. H. Chellai, 2010, Spatial and temporal distribution of ooids along a Jurassic carbonate ramp: Amellago outcrop transect, High-Atlas, Morocco: Geological Society, London, Special Publications, v. 329, no. 1, p. 65–88, doi:10.1144/SP329.4.

Response: Yes, carbonate debrite facies are found aroud the salt diapir adjacent to the rim syncline. Similar calcidebrites-turbidite horizons have not been reported elsewhere in Vigla limestone before, so we believe that has to been linked with salt diapirism. Of course basin subsidence can be controlled also by salt withdrawal since that is linked with salt expelled from the extended sub-basin axis by differential loading, but still this is a salt tectonic process than solely a tectonic process as is supported by previous models in the Ionian zone.

532-533   Remind the reader of your specific evidence for onlap (see comment on lines 233-236) and reference Fig. 11 for unconformities.

Response: Done.

Fig. 13. A speculative base salt horizon would be useful in panels 1-7

Response: We agree and this base horizon is included in the revised figure.

It would make the arguments easier to follow if there were an additional figure 14. It would show the stratigraphic column and nomenclature, the classic pre- syn- and post-rift divisions; the proposed divisions of [82] to [84] with timing of salt movement; and your proposed model with timing of extensional periods and of salt uplift and piercing.

Response: A new fig. 14 was added trying to address the most the reviewer’s valuable suggestion.

641        None of the references is in the (horrible) MDPI house standard, with the year between the italicized journal abbreviation and the volume number.

Response: References were homogenized in the same format.
